# X-ALMA: Plug & Play Modules and Adaptive Rejection for Quality Translation at Scale

**Haoran Xu**° **Kenton Murray**◁ **Philipp Koehn**◁ **Hieu Hoang**° **Akiko I. Eriguchi**° **Huda Khayrallah**◇,‡
°Microsoft, ◁Johns Hopkins University, ◇Amazon
`haoranxu@microsoft.com`, {`kenton, phi`}`@jhu.edu`,
{`hihoan,akikoe`}`@microsoft.com, hudakh@amazon.com`

## Abstract

Large language models (LLMs) have achieved remarkable success across various NLP tasks with a focus on English due to English-centric pre-training and limited multilingual data. In this work, we focus on the problem of translation, and while some multilingual LLMs claim to support for hundreds of languages, models often fail to provide high-quality responses for mid- and low-resource languages, leading to imbalanced performance heavily skewed in favor of high-resource languages. We introduce **X-ALMA**, a model designed to ensure top-tier performance across 50 diverse languages, regardless of their resource levels. X-ALMA surpasses state-of-the-art open-source multilingual LLMs, such as Aya-101 (Üstün et al., 2024) and Aya-23 (Aryabumi et al., 2024), in every single translation direction on the FLORES-200 and WMT'23 test datasets according to COMET-22. This is achieved by plug-and-play language-specific module architecture to prevent language conflicts during training and a carefully designed training regimen with novel optimization methods to maximize the translation performance. After the final stage of training regimen, our proposed **A**daptive-**R**ejection **P**reference **O**ptimization (**ARPO**) surpasses existing preference optimization methods in translation tasks.[1]

## 1 Introduction

Large language models (LLMs) such as the GPT series (Brown et al., 2020; OpenAI, 2023), Mistral (Jiang et al., 2023), LLaMA series (Touvron et al., 2023a;b; Dubey et al., 2024), Gemma series (Team et al., 2024a;b), *inter alia* have demonstrated impressive performance across various NLP tasks. However, the efficacy of LLMs has primarily been evaluated on English tasks, with their multilingual capabilities receiving less attention due to the models being predominantly pre-trained on English and the scarcity of multilingual data. Recently, there has been a shift towards multilingual studies in LLMs. For instance, LLaMA 3 and 3.1 (Dubey et al., 2024) expand the vocabulary from 32K to 128K and pre-train on multilingual texts; Üstün et al. (2024) have introduced Aya-101, a multilingual generative model supporting 101 languages; and BigTranslate (Yang et al., 2023) and LLaMAX (Lu et al., 2024) scale LLM-based multilingual translation models to over 100 languages.

Despite the increased language support in LLMs, their performance across most languages falls short of practical application expectations, especially for mid- and low-resource languages (*weakness 1*). Furthermore, the performance of high-resource languages tends to be inferior compared to LLMs trained with fewer languages, a phenomenon known as the 'curse of multilinguality' (Conneau et al., 2020) (*weakness 2*). The weaknesses are prevalent in most current state-of-the-art (SoTA) massively multilingual models: overall quality decreases as the number of supported languages increases. Although methods such as building models by focusing on a smaller number of high-resource languages like German and Chinese can achieve satisfactory performance for these languages and mitigate these weaknesses (Aryabumi et al., 2024; Xu et al., 2024a;b; Alves et al.,

---

‡Work done at Microsoft.

[1]Code is released at `https://github.com/fe1ixxu/ALMA`. Models and Dataset are released at `https://huggingface/X-ALMA`.

2024), they neglect the needs of mid- and low-resource languages. In this work, we address these weaknesses and build a multilingual model that achieves consistently high performance across 50 diverse languages, regardless of resource level, with a focus on multilingual machine translation.

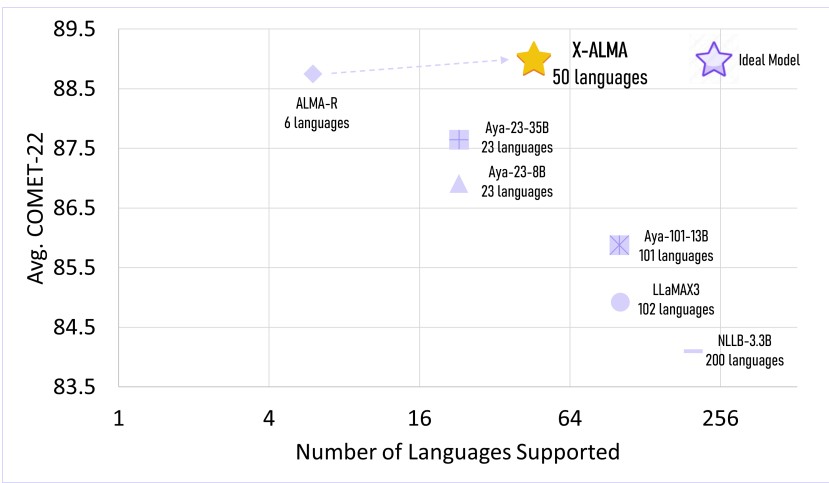

Figure 1: Depiction of the general inverse trend between the number of supported languages and average translation performance. While many state-of-the-art multilingual models claim to support hundreds of languages, the translation quality is not as high as in models trained on fewer languages, particularly for mid- and low-resource languages. This is reflected in the trend of decreasing average scores as more languages are supported. In contrast, we propose X-ALMA, which extends ALMA-R (Xu et al., 2024a;b) by supporting 44 additional diverse languages with even higher average performance, offering top performance across all supported languages, regardless of resource level.

To visualize these weaknesses, let us closely examine current models in the context of multilingual translation. We evaluate each model on the overlapping set of languages that are supported by the model and the 50 languages we focus on in this paper.[2] As shown in Figure 1, there is a clear trend: as the number of supported languages increases, the average translation performance decreases. This is intuitively understandable, as it is difficult for mid- and low-resource languages to reach the same level of performance as high-resource languages, thus lowering the overall average. For instance, ALMA-R (Xu et al., 2024b) achieves the highest average translation performance across the 6 languages it supports, while NLLB-200 (Team et al., 2022) exhibits the lowest average performance on 50 languages, largely due to poorer results in low-resource languages. Although this comparison is not entirely fair due to the varying number of languages tested, it provides a general indication of above-mentioned weaknesses in multilingual models.[3]

Despite the ability of current multilingual models to support hundreds of languages, the hollow purple star '☆' in the figure represents our ideal model, where the inclusion of more languages does not diminish the average performance. In this work, we introduce our multilingual translation model, **X-ALMA**, represented by the solid golden star '⭐' in Figure 1, which extends ALMA(-R) (Xu et al., 2024a;b) from 6 languages to 50 languages. ALMA-R is one of top-performing translation models built on LLMs, comparable to WMT winners and GPT-4-turbo. Despite the addition of 44 more languages, X-ALMA even achieves slightly higher average performance compared to ALMA-R.

We summarize our main contributions as follows, including our model architecture design and training methodology.

**Plug-and-Play Architecture:** For capacity reasons, we design X-ALMA with several different modules with each module serving a group of similar languages. These modules can either be plugged into the base model individually for the inference of target languages—reducing the neces-

---

[2]This is to depict a trend, and we acknowledge that scores are not directly comparable across languages.

[3]Here we evaluated these models using FLORES-200 (Team et al., 2022) test data and reporting the average COMET-22 (Rei et al., 2022) across all languages, to or from English.

sity of loading all expert parameters and saving memory—or all modules can be loaded together in a mixture-of-experts (MoE) way (Shazeer et al., 2017; Lepikhin et al., 2021).

**Effective Training Recipe:** The training regimen for X-ALMA consists of three pre-training stages and two post-training stages, each crucial for achieving optimal performance. Furthermore, in the final stage, we introduce **A**daptive-**R**ejection **P**reference **O**ptimization (**ARPO**), designed to maximize performance and address the 'over-rejection' issue (detailed explanation in Section 4) in translation preference learning, which current optimization methods struggle to resolve.

**State-of-the-Art Performance and Data Release:** X-ALMA outperforms existing open-source multilingual translation models *across 50 diverse languages for every single direction* only training on publicly available data, as measured by COMET-22. To enable future work, we also release the preference learning data for 50 languages and the model checkpoints.

## 2 BACKGROUND

### 2.1 PROBLEM DEFINITION

We consider a decoder-only LLM, denoted as $\pi_\theta$, parameterized by $\theta$, for multilingual machine translation tasks. Let $\mathcal{D}$ represent the multilingual dataset, consisting of pairs of a source sentence $x$ and the corresponding perfect translation $y$, represented as $\mathcal{D} = \{x, y\}$. Given a prompt $\mathcal{I}$ that instructs the model to perform the translation, our goal is to maximize the log-likelihood of the multilingual parallel dataset $\mathcal{D}$: $\max_\theta \ \mathbb{E}_{(x,y)\sim\mathcal{D}}[\log \pi_\theta(y|x, \mathcal{I})]$.

### 2.2 RELATED WORK

**Multilingual Translation** Massively Multilingual Translation models (Johnson et al., 2017), including open-source models such as PRISM Thompson & Post (2020a;b), M2M-100 (Fan et al., 2020), and NLLB (Team et al., 2022) combine translation between many language pairs in a single encoder-decoder model. T5 (Raffel et al., 2020) and mT5 (Xue et al., 2021) considered translation one of multitask learning.

**LLM-Based Translation** Initially, decoder-only LLMs struggled to match the performance of conventional encoder-decoder models for MT. For example, GPT-3.5 slightly under-performed the concurrent WMT winners (Hendy et al., 2023), and large open-source models like OPT-175B (Zhang et al., 2022) performed worse than the 1.3B parameter NLLB model (Team et al., 2022), even on high-resource languages, as demonstrated by Zhu et al. (2024b). This lead to an increased interest in smaller LLMs, such as 7B or 13B models, because even smaller models like NLLB-1.3B showed strong translation capabilities. However, first generation LLM-based MT models such as TIM (Zeng et al., 2023), SWIE (Chen et al., 2023), and BayLing (Zhang et al., 2023) still lag behind encoder-decoder models in performance. The under performance of LLMs on translation lead to hybrid approaches combining LLMs with dedicated NMT models (Petrick et al., 2023; Hoang et al., 2024). Recently, GPT-4 (OpenAI, 2023) has been reported to achieve top performance in the WMT competition (Kocmi et al., 2023), and smaller LLM-based models, like ALMA(-R) (Xu et al., 2024a;b) and Tower (Alves et al., 2024), have demonstrated comparable performance to GPT-4 by employing their specialized training methods. However, the high performance of LLM-based translation models is limited to a small subset of languages.

**Massively Multilingual LLM** The limited scope of languages in LLM-based MT models stems primarily from English-focused pre-training and the use of restricted vocabularies. However, this limitation has driven interest in expanding these models to support a broader range of languages. The simplest approach to extending current LLM-based MT models involves expanding the vocabulary and training on large amounts of parallel data across additional languages (Yang et al., 2023), but this approach has been shown to degrade model performance. Aya-101 (Üstün et al., 2024) revisits the encoder-decoder architecture, building a multilingual model based on the largest MT5 (Xue et al., 2020), designed not only for translation but also for general multilingual QA. Similarly, LLaMAX (Lu et al., 2024) extends LLaMA-2 and LLaMA-3 to over 100 languages. However, multilingual models often suffer from reduced performance on mid- and low-resource languages, which can also negatively impact high-resource language performance. To mitigate this, the decoder-only model Aya-23 (Aryabumi et al., 2024) focuses exclusively on 23 high-resource languages to maximize

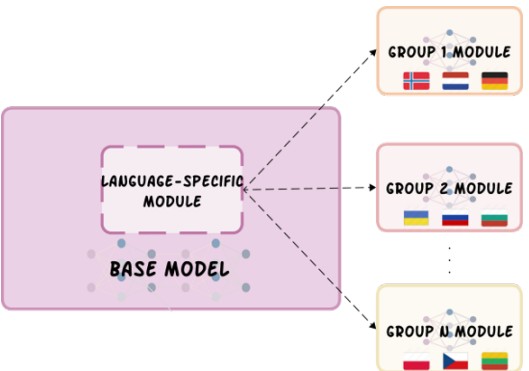

Figure 2: High-level architecture design of the plug-and-play multilingual model. Each language group is assigned a specific module that works alongside the base model. These language-specific modules handle inputs exclusively from their respective language groups, enabling the model to effectively adapt to different linguistic characteristics while leveraging the shared base model for comprehensive multilingual learning.

their performance and avoid the 'curse of multilinguality'. While limiting the number of supported languages can indeed alleviate some challenges, it reverses the goal of building truly multilingual models and neglects the needs of mid- and low-resource languages. In this paper, we expand ALMA-R from 6 languages to 50, ensuring robust performance across all languages.

## 3 METHODS

### 3.1 MODEL ARCHITECTURE

Our model architecture consists of: (1) a dense base model, and (2) multiple language-specific (LS) modules. The core concept of LS modules is to prevent conflicts between languages during training, such as gradient conflicts (Wang et al., 2021). This design has similarities with the mixture-of-experts (MoE) approach (Shazeer et al., 2017; Lepikhin et al., 2021), but diverges by not using a neural-based gate to assign tokens to experts (LS modules). Instead, similar to Xu et al. (2023), the assignment is hard-gated, i.e., input data is assigned exclusively to the module designated for its language. Consequently, only the base model and the corresponding LS module are activated, depending on the input language. Languages are categorized into distinct groups, with each group sharing a common LS module. An overview of the model architecture is illustrated in Figure 2.

In detail, the base model architecture is built upon the LLaMA-2 architecture (Touvron et al., 2023b). Each LS module comprises low-rank adaptations (LoRAs) (Hu et al., 2021) integrated into all linear layers within the attention and multi-layer perceptron (MLP) layers. The total number of parameters for each LS module is approximately 15% of the base model.

**Why This Design?** While model architectures such as MoE activate only one expert per example, all experts must still reside in GPU memory during training and inference, necessitating high-end GPUs. Moreover, MoE has been reported for its parameter inefficiency in multilingual settings, e.g., hard-gated language assignment can achieve similar performance to MoE while using 4 times fewer parameters (Xu et al., 2023). Compared to MoE, our design offers three distinct model-loading strategies for both training and inference: (1) selectively loading a single, on-demand LS module, which alleviates GPU memory constraints; (2) merging LS modules with the base model to generate a new LS LLM model that retains the same parameter count as the base model, facilitating subsequent use; and (3) loading both the base model and all LS modules as a larger, combined model, similar to the approach employed by MoE.

### 3.2 LANGUAGE GROUPING

In this paper, we consider a total of 50 languages, encompassing 14 scripts and 18 language families, to capture the linguistic diversity. The languages are categorized into 8 groups based on two

Table 1: Language grouping based on linguistic features and balanced number of languages.

| Group ID | Linguistic Feature | Languages |
|---|---|---|
| 1 | Germanic languages | af, da, de, is, nl, no, sv, (en) |
| 2 | Romance Languages | ca, es, gl, it, pt, ro, (en) |
| 3 | Eastern and Southern Slavic Languages | bg, mk, ru, sr, uk, (en) |
| 4 | Southeast Asian Languages | fr, id, mg, ms, th, vi, (en) |
| 5 | Central and Eastern European Languages | cs, el, hu, lt, lv, pl, (en) |
| 6 | Eurasian Language Mix | et, fi, ja, ka, ko, zh, (en) |
| 7 | Indo-Aryan Languages | gu, hi, mr, ne, ur, (en) |
| 8 | Turkic and Semitic Languages | ar, az, fa, he, kk, ky, tr, uz, (en) |

criteria: (1) each group should consist of languages that are as similar as possible, and (2) the number of languages in each group should be balanced. We opted not to use automated tools like Lang2Vec (Littell et al., 2017) for grouping, as we found that manual grouping based on human linguistic knowledge yields more accurate classification in line with our criteria. The specific languages within each group are presented in Table 1 with their `ISO-639-1` code. Note that English (`en`) is included in all groups to ensure that each group can perform English-centric translation. More detailed information on these languages is provided in Appendix A.

## 3.3 TRAINING RECIPE

We provide a comprehensive description of the training recipe for the X-ALMA model, including three pre-training stages and two post-training stages. An overview of this training recipe is depicted in the workflow diagram in Appendix B. The specifics of each stage are elaborated upon as follows.

**Pre-Training Stage 1: Monolingual Fine-Tuning Base Model** The first stage of pre-training is dedicated exclusively to the base model. During this phase, we fine-tune the base model using 20B monolingual tokens from all 50 languages, with a sampling ratio proportional to the size of the available monolingual data for each language, as suggested by Xu et al. (2024a). This stage aims to facilitate the model's acquisition of fundamental knowledge across all languages.

**Pre-Training Stage 2: Monolingual Fine-Tuning Language-Specific Modules** In all subsequent stages, the base model remains frozen, and the focus shifts to fine-tuning LS modules. During the second stage of pre-training, each LS module is fine-tuned with 10B monolingual tokens exclusively from the languages within its respective group. This stage is designed to enable each LS module to emphasize on learning general knowledge across the specific languages.

**Pre-Training Stage 3: Pseudo-Monolingual Fine-tuning** In this stage, we continue to fine-tune the LS modules using *pseudo-monolingual data* from each module's language group. This pseudo-monolingual data is constructed from parallel sentences. While previous studies have indicated that simple instruction tuning with a large volume of parallel sentences for instruction tuning can degrade model performance (Xu et al., 2024a; Zhu et al., 2024a), recent research demonstrates that utilizing parallel data in the pre-training stage can enhance multilingual alignment (Alves et al., 2024; Kondo et al., 2024; Lu et al., 2024). Similar to these approaches, we combine each available translation pair to create a new sentence in either a *<source sentence><target sentence>* or *<target sentence><source sentence>* manner, with the order of the source and target sentence in each pair determined randomly. We then concatenate all the combined translations to construct the pseudo-monolingual data. Each LS module is fine-tuned on 1.25B tokens.

**Post-Training Stage 1: Supervised Fine-tuning** Building on the insights from prior research that small but high-quality multilingual datasets are sufficient to yield impressive performance (Maillard et al., 2023; Xu et al., 2024a), we supervised fine-tune (SFT) the model using a small, high-quality parallel dataset at this stage with the translation prompt suggested by Xu et al. (2024a). This fine-tuning is performed using a simple causal language modeling (CLM) loss.

**Post-Training Stage 2: Preference Optimization** We also introduce *Adaptive Rejection Preference Optimization (ARPO)* to further enhance translation quality across all languages. ARPO is designed to address the 'over-rejection' issue in MT preference learning, a challenge that other pref-

erence optimization methods struggle to manage effectively. We will elaborate on our motivations, methodology, and preference data construction in the following section.

# 4 ADAPTIVE-REJECTION PREFERENCE OPTIMIZATION

## 4.1 LIMITATIONS IN CURRENT PREFERENCE LEARNING

When constructing preference data for MT, it is essential that the dis-preferred translation is also of high quality to ensure meaningful model improvement (Xu et al., 2024b). This results in a scenario where the preferred and dis-preferred translations are often very similar, differing by only a few words, which is quite different from the preference data used in open-ended question-answering (QA) tasks (A detailed example is shown in Appendix C). While many preference optimization methods have proven effective in various NLP tasks (Rafailov et al., 2024; Azar et al., 2024; Hong et al., 2024; Meng et al., 2024), we find that they are not well-suited for the MT task because they tend to reject the entire dis-preferred translation which is similar to the preferred one. This approach can inadvertently lead to the rejection of most tokens in the preferred translation as well, resulting in a phenomenon we term *over-rejection*, where the writing style of the translation outputs is forced away from the preferred data distribution (further analysis and examples in Section 6.1).

Mathematically speaking, the preference optimization problem can be generally formulated given a dataset $\mathcal{D} = \left\{ x^{(i)}, y_w^{(i)}, y_l^{(i)} \right\}_{i=1}^{N}$, where each data point consists of a prompt (source sentence) $x$, a preferred response (translation) $y_w$, and a dis-preferred response $y_l$, for a total of $N$ data points:

$$\mathcal{L} = \mathbb{E}_{(x,y_w,y_l)\sim\mathcal{D}}\Big[f\Big(r_\theta(y_w|x) - r_\theta(y_l|x)\Big)\Big], \tag{1}$$

where $f(\cdot) : \mathbb{R} \to \mathbb{R}$ is a general non-linear function. In many instances, such as in the DPO method (Rafailov et al., 2024), $f$ is the negative log-likelihood of the Bradley-Terry objective, i.e., $f(\cdot) = -\log\sigma(\cdot)$. Here, $r_\theta(y|x)$ represents the reward of $y$, calculated according to log-probability of the policy model parameterized by $\theta$. In the case of DPO, the reward function is defined as $r_\theta(y|x) = \beta\log(\pi_\theta(y|x)) - \beta\log(\pi_{\text{ref}}(y|x))$, where $\beta$ is a hyperparameter and $\pi_{\text{ref}}$ is the reference model.

As indicated by Equation 1, when $y_w$ and $y_l$ are too similar, the difference between $r_\theta(y_w|x)$ and $r_\theta(y_l|x)$ tends to be small and even near 0. Consequently, the near-zero difference between rewards causes the preference loss to a constant value, such as $f(0) = -\log\sigma(0)$ in the case of DPO. This makes it challenging for the optimization process to distinguish between the two options, hindering meaningful improvements.

The challenge becomes even more pronounced in a finite-data regime. While an infinite number of response pairs with small but precise preference differences could mitigate the optimization difficulties, translation preference data is often sparse and may contain noise (e.g., the AI-labeled preference data used by Xu et al. (2024b) contains only 2K samples per direction). Consequently, the model is prone to overfitting to these minor differences, which poses a significant empirical challenge and can lead to suboptimal learning outcomes, particularly when dealing with a large response (translation) space, as is the case with LLMs.

## 4.2 ADAPTIVE REJECTION

To mitigate the over-rejection, we introduce an adaptive penalty, denoted as $\tau_\theta$, which controls the strength of the dis-preferred term in the loss:

$$\mathcal{L}_{\text{ARPO}} = \mathbb{E}_{(x,y_w,y_l)\sim\mathcal{D}}\Big[f\Big(r_\theta(y_w|x) - \tau_\theta(y_w,y_l) \cdot r_\theta(y_l|x)\Big)\Big]. \tag{2}$$

The value of $\tau_\theta$ is determined by the similarity between $y_w$ and $y_l$, ranging from 0 to 1:

$$\tau_\theta(y_w,y_l) = \min(e^{\eta \cdot z_\theta(y_w,y_l)} - 1, 1), \tag{3}$$

where $\eta$ is a hyperparameter, and $z_\theta(y_w,y_l)$ is a function that quantifies the distance between the the preferred and dis-preferred responses by measuring absolute difference of their average log-likelihoods:

$$z_\theta(y_w,y_l) = \text{abs}\big(\frac{\log(\pi_\theta(y_w|x))}{|y_w|} - \frac{\log(\pi_\theta(y_l|x))}{|y_l|}\big), \tag{4}$$

When $y_w$ and $y_l$ are very similar, the absolute difference between their averaged log-likelihoods is small, resulting in $\tau_\theta$ close to 0, thereby reducing the impact of the dis-preferred term on the loss and mitigate rejection on this translation. Conversely, when the difference between $y_w$ and $y_l$ is large, $\tau_\theta$ close to 1, turning the loss back to a standard preference optimization loss.

In the multilingual MT task, we start with contrastive preference optimization (CPO) (Xu et al., 2024b), which has proven to be one of the most effective optimization methods for translation.

$$\mathcal{L}_{\text{CPO}} = -\mathbb{E}_{(x,y_w,y_l)\sim\mathcal{D}} \Big[ \underbrace{\log \sigma \Big( \beta \log \pi_\theta(y_w|x) - \beta \log \pi_\theta(y_l|x) \Big)}_{\text{preference loss}} + \underbrace{\log \pi_\theta(y_w|x)}_{\text{BC loss}} \Big]. \quad (5)$$

CPO consists of two components: preference loss and behavior cloning (BC) loss (Hejna et al., 2023). The BC loss helps prevent the model from drifting too far from the original task. Then, we incorporate adaptive rejection into the preference term in CPO, resulting in a new loss function:

$$\mathcal{L}_{\text{ARPO}} = -\mathbb{E}_{(x,y_w,y_l)\sim\mathcal{D}} \Big[ \log \sigma \Big( \beta \log \pi_\theta(y_w|x) - \tau_\theta(y_w, y_l) \cdot \beta \log \pi_\theta(y_l|x) \Big) + \log \pi_\theta(y_w|x) \Big]. \quad (6)$$

## 5 EXPERIMENTS

### 5.1 DATA

**Monolingual and Parallel Data**   Following the introduction of 50 languages in Section 3.2, we focus on 98 English-centric translation directions, both into and from English. We test on Flores-200 test data (Team et al., 2022) and WMT'23 (Kocmi et al., 2023). For pre-training stages 1 and 2, we use monolingual data from OSCAR (Ortiz Su'arez et al., 2019). In pre-training stage 3, we construct pseudo-monolingual data using NLLB (Schwenk et al., 2021; Heffernan et al., 2022; Team et al., 2022)[4] and OPUS (Tiedemann, 2012; Zhang et al., 2020) parallel training data. Web crawled data (included in NLLB) has been shown to contain substantial mis-aligned and mis-translated segments (Khayrallah & Koehn, 2018; Kreutzer et al., 2022) and low-quality machine translated segments (Thompson et al., 2024). Therefore, in the SFT step—building on the insights from Xu et al. (2024a) that a small amount of *high-quality* data can significantly enhance translation performance—we use the Flores-200 dev set and NTREX (Barrault et al., 2019; Federmann et al., 2022) test data as our training data to ensure the quality. Given that both Flores-200 and NTREX are multi-way-parallel datasets (all languages share the same English source sentences), we also incorporate the WMT'15-22 test data in training. The final data size in the SFT stage for each direction ranges from 3K to 7K, with an average of 4K per direction.

**Preference Data Construction**   Given the scarcity of preference datasets for multilingual MT, we describe our approach to constructing preference data for 50 languages. Starting with the parallel data used in SFT, for each source sentence $x$, we generate a translation $y_{\text{xalma}}$ using X-ALMA that has been fine-tuned through SFT. Then, the reference translation $y_{\text{ref}}$ is designated as the preferred translation, and $y_{\text{xalma}}$ as the dis-preferred one, forming our initial preference dataset, denoted as $\mathcal{D}_1 = \{x, y_{\text{ref}}, y_{\text{xalma}}\}$. Unlike Xu et al. (2024b), we avoid the use of reference-free methods like XCOMET (Guerreiro et al., 2023) for ranking translations in preference data construction to avoid potential bias, as the same metrics are used for evaluation. As a result, $\mathcal{D}1$ might contain some noise due to the assumption that reference translations are always preferred. To reduce this noise, for high-resource languages, we also employ GPT-4o to produce revised translations $y_{\text{gpt}}$ conditioned on $(x, y_{\text{xalma}})$, drawing on studies that show post-editing by LLMs can improve translation quality (Ki & Carpuat, 2024; Feng et al., 2024; Raunak et al., 2023). We show the prompts in Appendix D. Thus, our second preference dataset is defined as $\mathcal{D}_2 = \{x, y_{\text{gpt}}, y_{\text{xalma}}\}$. We then concatenate the two datasets to form the final preference dataset, denoted as $\mathcal{D} = \mathcal{D}_1 \cup \mathcal{D}_2$.

### 5.2 EVALUATION

We report COMET-22 (Rei et al., 2022) as our main metric as suggested by Freitag et al. (2023; 2024). In Appendix E, we also include XCOMET-XL (without references) (Guerreiro et al., 2023) as recommended by Xu et al. (2024b), and BLEU (Papineni et al., 2002; Post, 2018) for completeness.

---

[4]https://huggingface.co/datasets/allenai/nllb

Table 2: The overall results of Flores test data across each language group in en→xx. Scores are reported using COMET-22. X-ALMA outperforms both massively multilingual models, such as Aya-101, and models focus specifically on high-resource languages, like Aya-23. 'All' represents the average performance across all languages in the group, while 'High' refers to the average performance for high-resource languages in the group. **Bold numbers** represent the highest scores.

| Models | Group 1 | | Group 2 | | Group 3 | | Group 4 | | Group 5 | | Group 6 | | Group 7 | | Group 8 | |
|---|---|---|---|---|---|---|---|---|---|---|---|---|---|---|---|---|
| | All | High | All | High | All | High | All | High | All | High | All | High | All | High | All | High |
| LLaMA-3.1-8B-Instruct | 80.8 | 79.8 | 83.7 | 84.2 | 79.1 | 69.2 | 76.3 | 85.8 | 79.0 | 81.2 | 71.1 | 71.8 | 70.1 | 69.6 | 78.3 | 84.4 |
| NLLB-3.3B | 88.2 | 88.8 | 88.3 | 88.1 | 89.4 | 89.1 | 87.1 | 88.2 | 89.2 | 89.8 | 87.5 | 87.5 | 80.1 | 80.9 | 88.1 | 87.5 |
| LLaMAX3-Alpaca-8B | 86.4 | 86.9 | 86.8 | 86.6 | 85.7 | 82.0 | 81.7 | 86.2 | 86.6 | 87.1 | 86.0 | 87.3 | 76.5 | 76.6 | 82.6 | 83.6 |
| Aya-101 | 85.0 | 85.7 | 86.8 | 86.2 | 87.7 | 85.6 | 85.8 | 85.5 | 88.4 | 88.7 | 87.5 | 87.3 | 76.2 | 75.5 | 86.8 | 86.3 |
| Aya-23-8B | 75.1 | 84.7 | 86.6 | 86.6 | 74.4 | 75.7 | 74.6 | 88.7 | 70.6 | 77.3 | 67.1 | 79.8 | 68.9 | 79.3 | 76.0 | 87.9 |
| Aya-23-35B | 79.6 | 86.5 | 87.1 | 87.0 | 77.6 | 78.5 | 76.7 | 88.6 | 82.1 | 86.0 | 73.9 | 84.4 | 61.9 | 79.1 | 68.8 | 87.8 |
| X-ALMA (only SFT) | 89.5 | 89.7 | 89.2 | 88.9 | 90.7 | 90.2 | 88.1 | 89.1 | 90.6 | 90.7 | 90.1 | 90.4 | 82.6 | 81.4 | 89.2 | 88.9 |
| X-ALMA | **89.6** | **89.9** | **89.4** | **89.0** | **90.9** | **90.5** | **88.6** | **89.5** | **91.0** | **91.1** | **90.6** | **90.8** | **83.2** | **81.9** | **89.4** | **89.2** |

Table 3: The overall COMET-22 scores of Flores test data across each language group in xx→en. Similarly, X-ALMA outperforms all baselines.

| Models | Group 1 | | Group 2 | | Group 3 | | Group 4 | | Group 5 | | Group 6 | | Group 7 | | Group 8 | |
|---|---|---|---|---|---|---|---|---|---|---|---|---|---|---|---|---|
| | All | High | All | High | All | High | All | High | All | High | All | High | All | High | All | High |
| LLaMA-3.1-8B-Instruct | 68.8 | 77.6 | 70.9 | 76.9 | 51.2 | 53.8 | 65.6 | 76.4 | 54.8 | 60.2 | 58.8 | 66.9 | 47.6 | 53.7 | 53.7 | 67.5 |
| NLLB-3.3B | 79.1 | 81.8 | 84.5 | 85.0 | 84.3 | 83.8 | 81.1 | 85.4 | 74.9 | 76.0 | 76.1 | 77.3 | 88.3 | 88.9 | 79.5 | 81.6 |
| LLaMAX3-Alpaca-8B | 88.3 | 88.5 | 88.1 | 87.9 | 87.0 | 86.8 | 86.2 | 87.9 | 87.0 | 87.2 | 81.7 | 87.7 | 83.6 | 88.9 | 84.7 | 87.7 |
| Aya-101 | 87.2 | 88.2 | 87.6 | 87.6 | 85.4 | 85.5 | 86.2 | 87.6 | 86.3 | 86.5 | 86.5 | 86.7 | 84.8 | 87.5 | 85.9 | 87.1 |
| Aya-23-8B | 84.6 | 88.2 | 87.9 | 87.7 | 83.3 | 83.3 | 79.8 | 88.5 | 82.9 | 85.2 | 79.9 | 86.1 | 71.8 | 89.1 | 76.7 | 88.0 |
| Aya-23-35B | 87.4 | 88.9 | 88.8 | 88.6 | 86.3 | 86.2 | 82.3 | 88.7 | 86.4 | 87.2 | 85.9 | 88.0 | 79.4 | 89.6 | 82.7 | 88.6 |
| X-ALMA (only SFT) | 89.1 | 89.2 | 88.8 | 88.6 | 87.9 | 87.7 | 87.7 | 88.8 | 87.9 | 88.1 | 88.2 | 88.3 | 89.3 | 89.8 | 87.5 | 88.4 |
| X-ALMA | **89.4** | **89.5** | **89.2** | **89.0** | **88.1** | **87.8** | **88.0** | **88.9** | **88.2** | **88.4** | **88.7** | **88.8** | **89.6** | **90.1** | **88.0** | **88.9** |

## 5.3 TRAINING SETUP

We use ALMA-13B-Pretrain (Xu et al., 2024a) as our backbone model, which is pre-trained on 6 languages and based on LLaMA-2 (Touvron et al., 2023b). Following Xu et al. (2024a), we pre-train the backbone model with a batch size of 256, a warm-up ratio of 0.01, and sequences containing up to 512 tokens. In the post-training stage, the model is fine-tuned for many-to-many multilingual translation manner using 1 epoch with a batch size of 128, and other settings remain unchanged. For preference learning, we set $\eta$ as 1.5 and $\beta$ as 0.1 for all experiments.

## 5.4 BASELINES

We use the strongest open-source massively multilingual translation models as our baselines, including NLLB-200 (Team et al., 2022), Aya-101 (Üstün et al., 2024), and LLaMAX3-Alpaca (Lu et al., 2024). Additionally, we compare our model's translation performance with Aya-23-8B and Aya-35B (Aryabumi et al., 2024) to demonstrate that increasing the number of supported languages does not compromise the performance of high-resource languages, effectively mitigating the curse of multilinguality. We also include LLaMA-3.1-8B-Instruct as a baseline to assess the performance of one of the latest strong LLMs in multilingual translation.

## 5.5 RESULTS

We present the average performance for each language group in both en→xx and xx→en directions on the Flores-200 test data in Tables 2 and 3. The results for WMT'23 in both directions are provided in Table 4. Detailed results for each translation direction can also be found in Appendix E.

**Compared with SoTA Multilingual Open Models:** General instruction-tuned LLaMA-3.1 significantly lags behind models specifically designed for translation, so we primarily focus on other models. X-ALMA outperforms other massively multilingual models such as NLLB-3.3B, LLaMAX3-Alpaca-8B, and Aya-101 on average across all language groups, both into and from English for both Flores-200 and WMT'23 test sets. Furthermore, X-ALMA surpasses Aya-23-8B and Aya-23-35B—both of which are tailored for high-resource languages—on average across all high-resource languages in each group. In fact, as detailed in Appendix E, *X-ALMA surpasses all baselines in all translation directions according to COMET-22 and outperforms in 97 out of 98 directions based on XCOMET-XL*, achieving top translation performance for all languages considered.

Table 4: Results on WMT'23 dataset reported using COMET-22. The symbol → represents translations from English into the target language, while ← indicates translations into English.

| Models | de → | de ← | zh → | zh ← | ja → | ja ← | ru → | ru ← | uk → | uk ← | he → | he ← | Avg. → | Avg. ← |
|---|---|---|---|---|---|---|---|---|---|---|---|---|---|---|
| ALMA-R-13B | 84.0 | 85.5 | 85.0 | 80.6 | - | - | 85.5 | **83.3** | - | - | - | - | - | - |
| TowerInstruct-7B-v0.2 | 83.1 | 84.6 | 85.6 | 80.5 | - | - | 85.3 | 83.1 | - | - | - | - | - | - |
| NLLB-3.3B | 79.7 | 66.6 | 79.6 | 67.8 | 81.6 | 65.8 | 83.8 | 76.7 | 82.8 | 79.0 | 83.6 | 79.9 | 81.8 | 72.6 |
| LlamaX3-8B | 73.3 | 79.4 | 81.5 | 79.3 | 81.8 | 80.1 | 81.6 | 81.3 | 80.6 | 84.9 | 82.5 | 83.0 | 80.2 | 81.3 |
| Aya-101 | 75.1 | 81.6 | 78.6 | 73.7 | 84.6 | 77.3 | 83.1 | 81.4 | 82.7 | 84.5 | 82.0 | 82.9 | 81.0 | 80.2 |
| Aya-23-8B | 80.4 | 82.1 | 85.3 | 78.8 | 86.5 | 80.2 | 84.3 | 81.6 | 84.3 | 85.0 | 84.3 | 84.9 | 84.2 | 82.1 |
| Aya-23-35B | 80.7 | 82.3 | 84.6 | 79.7 | 86.4 | 81.6 | 84.7 | 82.2 | 84.0 | 85.7 | 84.1 | 85.9 | 84.1 | 82.9 |
| X-ALMA (only SFT) | 84.1 | 85.3 | 86.1 | 80.3 | 86.8 | 81.6 | 85.9 | 82.4 | 85.3 | 86.4 | 86.1 | 84.4 | 85.7 | 83.4 |
| X-ALMA | **84.4** | **85.7** | **86.7** | **80.9** | **87.5** | **82.4** | **86.3** | 83.3 | **85.5** | **86.8** | **86.2** | **85.6** | **86.1** | **84.1** |

Table 5: Average performance comparison of various preference optimization methods for en→xx and xx→en on Group 6.

| Models | Avg. en→xx BLEU | Avg. en→xx COMET-22 | Avg. en→xx XCOMET-XL | Avg. xx→en BLEU | Avg. xx→en COMET-22 | Avg. xx→en XCOMET-XL |
|---|---|---|---|---|---|---|
| XALMA (only SFT) | 26.5 | 90.1 | 80.3 | 32.1 | 88.2 | 77.4 |
| + DPO | 0.7 | 53.6 | 51.1 | 7.1 | 79.2 | 64.6 |
| + BC | 23.5 | 90.2 | 80.0 | 27.8 | 87.5 | 77.4 |
| + SimPO | 0.0 | 16.7 | 1.5 | 0.0 | 16.4 | 8.9 |
| + BC | 23.3 | 89.7 | 78.7 | 26.6 | 87.1 | 76.5 |
| + KTO | 22.1 | 89.8 | 79.2 | 26.4 | 87.1 | 76.5 |
| + BC | 26.4 | 90.3 | 80.4 | 29.2 | 87.5 | 77.1 |
| + ORPO | 23.0 | 85.8 | 75.9 | 22.7 | 81.8 | 70.8 |
| + CPO | 22.2 | 90.2 | 79.9 | 26.5 | 87.8 | 77.0 |
| + ARPO (Final X-ALMA) | **27.8** | **90.6** | **81.3** | **32.2** | **88.7** | **78.4** |

**Effectiveness of ARPO:** ARPO delivers consistent improvements compared to SFT-only models in Flores-200 and WMT'23. Similarly, as shown in the full results in Appendix E, *ARPO enhances performance in every translation direction as measured by COMET-22 and delivers improvements in 95 out of 98 directions according to XCOMET-XL.* We also compare the effectiveness of ARPO against other preference optimization methods in Section 6.1.

# 6 ANALYSIS

All analyses are conducted on languages in Group 6, as it is the most challenging group to learn due to its mix of typologically diverse Asian and European languages.

## 6.1 PREFERENCE OPTIMIZATION COMPARISON

Here, we compare ARPO with other popular optimization methods, including DPO (Rafailov et al., 2024), KTO (Ethayarajh et al., 2024), ORPO (Hong et al., 2024), SimPO (Meng et al., 2024), and the original CPO (Xu et al., 2024b). As indicated by CPO findings, directly applying preference learning to the MT task can harm the model, but adding a behavior cloning (BC) regularizer can stabilize training and improve the performance (Xu et al., 2024b). Following them, we also incorporate a BC regularizer into optimization methods that do not originally include it to provide a fair comparison. Table 5 presents the comparison of preference optimization methods across all three metrics. As shown, ARPO clearly outperforms all baselines.

**Over-Rejection** Over-rejection manifests itself under the significant shift in writing style away from the preferred data distribution. We observe a clear and big BLEU scores drop across all other preference optimization methods, indicating a decline in lexical matching. However, for certain methods, such as DPO + BC, KTO + BC, and CPO, both COMET scores do not decrease as drastically (and in some cases even improve slightly for en→xx), suggesting that the models still produce accurate translations that maintain the same semantic meaning, but with a different writing style. Some translation examples generated by CPO are provided in Appendix F. Unlike ARPO, methods such as CPO tend to produce a wider range of writing styles to convey the same meaning

as the reference, many of which are accurate and non-detrimental. However, excessive shifts in writing style still can introduce translation errors that negatively impact overall quality. These small number of errors, concealed within a lot of stylistic deviations, are where over-rejection occurs. As hypothesized in Section 4, the significant style shift is caused by the model rejecting dis-preferred translations that are similar to preferred ones, leading to an excessive rejection of certain writing styles from the preferred data. However, ARPO addresses this issue by *constraining the stylistic variation within a more controlled range*, thereby mitigating errors caused by over-rejection.

For other methods such as naive DPO and SimPO, which even though work well in other NLP tasks, the over-rejection severely impairs the model's ability to generate meaningful translations. The introduction of ARPO significantly mitigate the over-rejection issue (stable BLEU scores) and maximize the translation qualities (the highest scores in two COMET metrics).

## 6.2 ABLATION STUDY

**Training Recipe**   We investigate the impact of each step in the training recipe on model performance. The average results for Group 6, both into and from English, are presented in the left part of Figure 3. The results show a clear trend of consistent performance improvement with each step in the training process. Note that 'None' is the initial checkpoint in our recipe, ALMA-13B-Pretrain.

**Parallel Data for SFT**   For SFT, we use high-quality parallel data from three sources: NTREX, WMT, and the Flores-200 dev set. Here, we investigate how combining parallel datasets affects performance *during the SFT stage*. As shown on the right of Figure 3, using only NTREX data already achieves impressive average translation performance for Group 6. Adding high-quality WMT data further boosts average performance, particularly for translations into English data. We hypothesize that this improvement stems from the increased diversity of English data, which mitigates overfitting to the NTREX English domain—a known issue with multi-way-parallel data, as observed by Aharoni et al. (2019). Conversely, incorporating more Flores-200 dev data (also multi-way-parallel) into training does not result in significant gains, also suggesting that the strong translation performance is not driven by in-domain Flores-200 data.

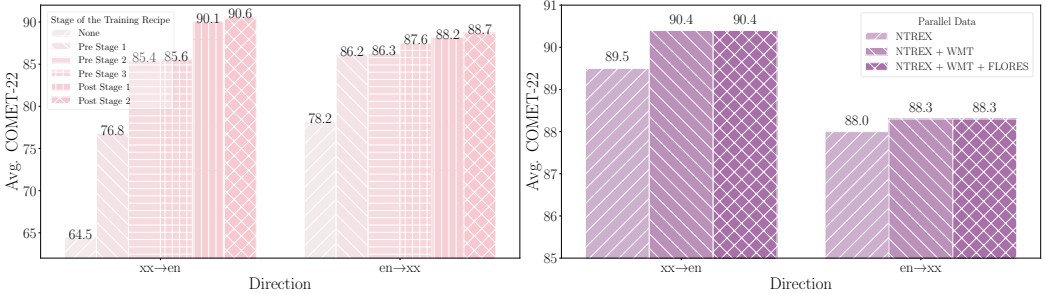

Figure 3: **Left:** ablation study on each stage of the training recipe, demonstrating that adding each stage leads to consistent performance improvements. **Right:** ablation study on the impact of parallel data composition during the SFT stage. Adding WMT data to NTREX significantly enhances model performance, while adding Flores-200 data provides no noticeable improvement.

## 7 CONCLUSION

We tackled the challenge of achieving high translation quality while scaling to a large number of languages, a limitation seen in many state-of-the-art multilingual models. We have introduced X-ALMA, an LLM-based multilingual translation system that prioritizes translation quality across all supported 50 languages, regardless of resource level. X-ALMA surpasses SoTA open models such as Aya-101 and Aya-23 in all translation directions on the FLORES-200 and WMT'23 test datasets, as measured by COMET-22. X-ALMA is built on a plug-and-play architecture with language-specific modules, complemented by a carefully designed training recipe. In particular, the final stage of the recipe, ARPO, achieves further performance gains and outperforms existing preference optimization methods in translation tasks, while successfully mitigating the over-rejection issue.

ACKNOWLEDGEMENTS

We thank anonymous reviewers for their insightful feedback. We express our profound appreciation to HyoJung Han, Jack Zhang, Tianjian Li, Thamme Gowda, Tom Kocmi, Young Jin Kim, Hany Hassan Awadalla, Marcin Junczys-Dowmunt, Vikas Raunak, Matt Post, Anoop Kunchukuttan, Roman Grundkiewicz, Arul Menezes, and Vishal Chowdhary for their engaging and valuable discussions that greatly enriched our work.

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

APPENDIX CONTENTS

## A  LANGUAGE INFORMATION

We provide detailed information on the eight language groups, including their scripts, language families, and resource levels, in Table 6. Each group includes English to ensure that each language-specific module supports English-centric translation and to prevent catastrophic forgetting of English. While we primarily grouped languages based on linguistic similarity, the grouping is not perfect. This is due to the need to balance the number of languages in each group and the inherent nature of language resources. For example, Group 6 is a mix of Asian and European languages, and although most languages in Group 4 are Southeast Asian languages, we include French as an additional bonus language to facilitate cross-lingual transfer, especially since most languages in this group are low- and mid-resource.

Table 6: Detailed information of all langauges.

| Language | ISO-639-1 | Script | Family | Subgroup | Resource |
|---|---|---|---|---|---|
| English | en | Latin | Indo-European | Germanic | High |
| *Group 1: Germanic languages* | | | | | |
| Afrikaans | af | Latin | Indo-European | Germanic | Mid |
| Danish | da | Latin | Indo-European | Germanic | Mid |
| Dutch | nl | Latin | Indo-European | Germanic | High |
| German | de | Latin | Indo-European | Germanic | High |
| Icelandic | is | Latin | Indo-European | Germanic | Low |
| Norwegian | no | Latin | Indo-European | Germanic | Low |
| Swedish | sv | Latin | Indo-European | Germanic | High |
| *Group 2: Romance Languages* | | | | | |
| Catalan | ca | Latin | Indo-European | Italic | High |
| Galician | gl | Latin | Indo-European | Italic | Mid |
| Italian | it | Latin | Indo-European | Italic | High |
| Portuguese | pt | Latin | Indo-European | Italic | High |
| Romanian | ro | Latin | Indo-European | Italic | Mid |
| Spanish | es | Latin | Indo-European | Italic | High |
| *Group 3: Eastern and Southern Slavic Languages* | | | | | |
| Bulgarian | bg | Cyrillic | Indo-European | Balto-Slavic | Mid |
| Macedonian | mk | Cyrillic | Indo-European | Balto-Slavic | Low |
| Russian | ru | Cyrillic | Indo-European | Balto-Slavic | High |
| Serbian | sr | Cyrillic | Indo-European | Balto-Slavic | High |
| Ukrainian | uk | Cyrillic | Indo-European | Balto-Slavic | Mid |
| *Group 4: Southeast Asian Languages* | | | | | |
| French | fr | Latin | Indo-European | Italic | High |
| Indonesian | id | Latin | Austronesian | Malayo-Polynesian | Mid |
| Malagasy | mg | Latin | Austronesian | Malayo-Polynesian | Low |
| Malay | ms | Latin | Austronesian | Malayo-Polynesian | Mid |
| Thai | th | Thai | Tai-Kadai | Kam-Tai | Mid |
| Vietnamese | vi | Latin | Austronesian | Vietic | High |

| Language | ISO-639-1 | Script | Family | Subgroup | Resource |
|---|---|---|---|---|---|
| *Group 5: Central and Eastern European Languages* | | | | | |
| Czech | cs | Latin | Indo-European | Balto-Slavic | High |
| Greek | el | Greek | Indo-European | Graeco-Phrygian | Mid |
| Hungarian | hu | Latin | Uralic | Finnic | High |
| Latvian | lv | Latin | Indo-European | Balto-Slavic | Mid |
| Lithuanian | lt | Latin | Indo-European | Balto-Slavic | Mid |
| Polish | pl | Latin | Indo-European | Balto-Slavic | High |
| *Group 6: Eurasian Language Mix* | | | | | |
| Chinese | zh | Han | Sino-Tibetan | Sinitic | High |
| Estonian | et | Latin | Uralic | Finnic | Mid |
| Finnish | fi | Latin | Uralic | Finnic | High |
| Georgian | ka | Georgian | Kartvelian | Georgian-Zan | Mid |
| Japanese | ja | Japanese | Japonic | Japanesic | High |
| Korean | ko | Hangul | Koreanic | Korean | High |
| *Group 7: Indo-Aryan Languages* | | | | | |
| Gujarati | gu | Gujarati | Indo-European | Indo-Aryan | Low |
| Hindi | hi | Devanagari | Indo-European | Indo-Aryan | High |
| Marathi | mr | Devanagari | Indo-European | Indo-Aryan | Low |
| Nepali | ne | Devanagari | Indo-European | Indo-Aryan | Low |
| Urdu | ur | Arabic | Indo-European | Indo-Aryan | Mid |
| *Group 8: Turkic and Semitic Languages* | | | | | |
| Arabic | ar | Arabic | Afro-Asiatic | Semitic | High |
| Azerbaijani | az | Arabic/Latin | Turkic | Common Turkic | Low |
| Hebrew | he | Hebrew | Afro-Asiatic | Semitic | Mid |
| Kazakh | kk | Cyrillic | Turkic | Common Turkic | Mid |
| Kyrgyz | ky | Cyrillic | Turkic | Common Turkic | Low |
| Persian | fa | Arabic | Indo-European | Iranian | High |
| Turkish | tr | Latin | Turkic | Common Turkic | High |
| Uzbek | uz | Latin | Turkic | Common Turkic | Low |

## B  ILLUSTRATION OF TRAINING RECIPE

Here, we illustrate an overview of our 5-step training recipe in Figure 4.

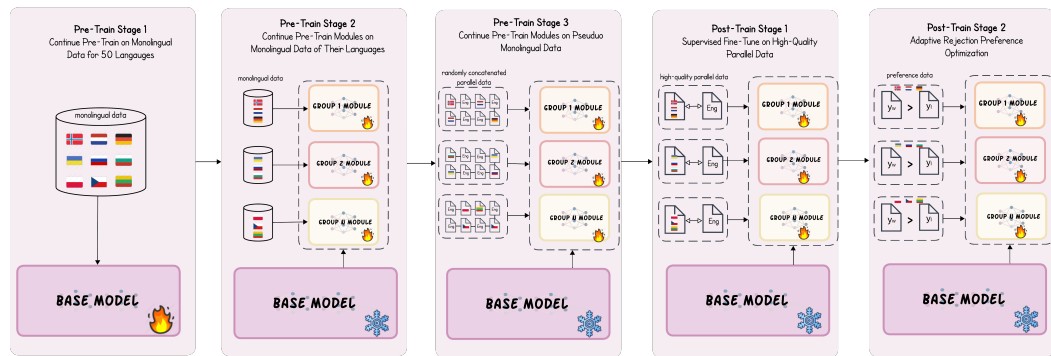

Figure 4:  This diagram of the multi-stage process of fine-tuning a multilingual model. *In Pre-Training Stage 1*, the base model is fine-tuned using 20B tokens of monolingual data from 50 languages. The process continues with *Pre-Training Stage 2*, where language-specific modules are fine-tuned with 10B monolingual tokens. *Pre-Training Stage 3* introduces pseudo-monolingual fine-tuning, using randomly concatenated parallel sentences to improve multilingual alignment. The model then undergoes *Post-Training Stage 1*, where SFT is performed on high-quality parallel data, followed by *Post-Training Stage 2*, which applies Adaptive Contrastive Preference Optimization to address over-rejection issues in translation preference learning.

## C    REWARD DIFFERENCES OF MT AND QA

Here, we present a comparison of the reward difference between machine translation (MT) tasks and open-ended question answering (QA) tasks in preference learning. Figure 5 illustrates the cumulative distribution of reward differences for the MT preference dataset, as described in Section 5, alongside the multilingual preference data from the Aya open-ended QA dataset (Singh et al., 2024) for languages in Group 6. The reward differences are sorted in ascending order, and their cumulative probabilities are displayed. The reward difference is computed using the CPO loss function: $\log \pi_\theta(y_w|x) - \log \pi_\theta(y_l|x)$. The construction of the Aya preference dataset follows the same methodology as the MT preference data, where we fine-tune the Aya QA dataset via SFT and use the fine-tuned model to generate answers for the training data. System-generated responses are treated as dis-preferred, while original references are considered preferred. As shown in Figure 5, the open-ended QA task exhibits significantly larger reward differences compared to machine translation. For instance, the maximum reward difference for the smallest 80% of MT preference data is 20, whereas it is approximately 300 for Aya QA. Similarly, the maximum reward difference for the MT preference data is 131, while that for Aya QA is nearly tenfold larger.

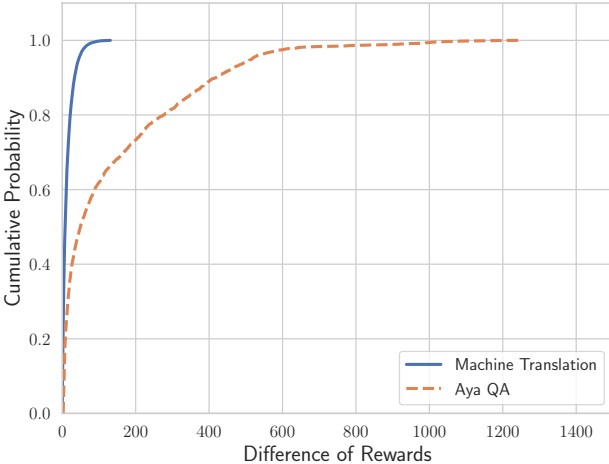

Figure 5:   Cumulative distribution of reward differences between machine translation and open-ended question answering tasks in contrastive preference optimization.

## D    PROMPTS

In Figure 6, we present the prompt used for GPT-4o post-editing during the construction of the preference dataset, as well as the prompt used for X-ALMA in generating translations.

## E    FULL RESULTS

We report translation quality using XCOMET-XL, as recent WMT metric shared tasks (Freitag et al., 2023; 2024) have found high correlation between trained metrics like XCOMET-XL and human preferences. However, those findings are limited to a few languages, and correlation with human judgments has also been shown to degrade for trained metrics in out of domain (relative to WMT, i.e. FLORES) settings (Zouhar et al., 2024). For these reasons we also report the more traditional BLEU metric.

Tables 7 to 14 present the results for each translation direction across language groups in the Flores-200 dataset, while Table 15 shows the full results for the WMT'23 dataset. On the Flores-200 dataset, X-ALMA surpasses all other open-source multilingual models in every translation direction according to COMET-22, and in 97 out of 98 directions according to XCOMET-XL. Additionally, ARPO, when compared to SFT, demonstrates superior performance in all translation directions reported by COMET-22 and in 95 out of 98 directions according to XCOMET-XL.

---

**GPT-4o Post-Edit Prompt**

**System**:
You are a native speaker of both *<source language>* and *<target language>*. You are an expert post editor of translations from *<source language>* into *<target language>* and a helpful assistant dedicated to improving translation quality. You will be provided with a source sentence in *<source language>* and its translation in *<target language>*. Your task is to carefully analyze provided source sentence and translation, and suggest improvements to the translation. Note that you only need to generate a refined translation in *<target sentence>* and do not generate anything else.

**User:**
The source sentence in *<source language>* is: *<source sentence>*
The translation in *<target language>* is: *<X-ALMA translated sentence>*
Note that you only need to generate a refined translation in *<target language>* and do not generate anything else.

---------------------------------------------------------------------------------------------------------------------

**X-ALMA Translation Chat Template**

[INST] Translate this from *<source language>* to *<target language>*:
*<source language>*: *<source sentence>*
*<target language>*: [/INST]

---

Figure 6: Prompts used for GPT-4o post editing and X-ALMA translation generation.

Table 7: Full results for Group 1 in the Flores test data.

| Models | en→af BLEU | COMET-22 | XCOMET | en→da BLEU | COMET-22 | XCOMET | en→de BLEU | COMET-22 | XCOMET | en→is BLEU | COMET-22 | XCOMET |
|---|---|---|---|---|---|---|---|---|---|---|---|---|
| LLaMA-3.1-8B-Instruct | 34.4 | 84.8 | 74.5 | 37.3 | 86.6 | 71.7 | 30.2 | 77.5 | 68.8 | 11.9 | 69.4 | 53.6 |
| NLLB-3.3B | 38.9 | 87.4 | 74.8 | 44.5 | 90.0 | 76.8 | 40.0 | 88.1 | 76.2 | 24.5 | 84.6 | 74.1 |
| LLaMAX2-Alpaca-7B | 38.1 | 86.2 | 73.2 | 40.3 | 89.5 | 76.2 | 32.2 | 86.6 | 75.0 | 20.4 | 82.9 | 72.3 |
| LLaMAX3-Alpaca-8B | 38.5 | 86.0 | 72.7 | 38.2 | 88.6 | 73.6 | 31.4 | 85.4 | 72.4 | 18.3 | 81.2 | 69.5 |
| Aya-101 | 22.5 | 78.8 | 40.8 | 34.2 | 87.6 | 62.9 | 29.3 | 84.3 | 67.5 | 20.9 | 84.3 | 74.5 |
| Aya-23-8B | 17.6 | 79.6 | 68.2 | 19.3 | 76.4 | 56.6 | 36.8 | 88.1 | 77.0 | 1.6 | 38.4 | 9.6 |
| Aya-23-35B | 26.7 | 81.2 | 67.9 | 29.0 | 82.9 | 65.5 | 37.0 | 88.1 | 77.2 | 5.9 | 51.0 | 28.6 |
| X-ALMA (only SFT) | **44.2** | 87.5 | 75.0 | 48.6 | 91.8 | 79.2 | **41.2** | 88.7 | 77.9 | **28.0** | **87.2** | **78.5** |
| X-ALMA (Ours) | 43.0 | **87.6** | **75.8** | **48.9** | **92.0** | **79.7** | 41.1 | **88.8** | **78.0** | 27.4 | **87.2** | **78.5** |

| Models | en→nl BLEU | COMET-22 | XCOMET | en→no BLEU | COMET-22 | XCOMET | en→sv BLEU | COMET-22 | XCOMET | Avg. en→xx BLEU | COMET-22 | XCOMET |
|---|---|---|---|---|---|---|---|---|---|---|---|---|
| LLaMA-3.1-8B-Instruct | 22.1 | 81.1 | 72.5 | 27.0 | 85.6 | 74.0 | 34.4 | 80.8 | 67.3 | 28.2 | 80.8 | 68.9 |
| NLLB-3.3B | 27.5 | 87.5 | 76.7 | 33.0 | 88.9 | 76.6 | 44.3 | 90.7 | 78.0 | 36.1 | 88.2 | 76.2 |
| LLaMAX2-Alpaca-7B | 23.4 | 86.4 | 76.2 | 30.1 | 88.9 | 78.1 | 39.3 | 89.6 | 77.7 | 32.0 | 87.1 | 75.5 |
| LLaMAX3-Alpaca-8B | 23.3 | 86.3 | 75.9 | 28.0 | 87.8 | 74.2 | 38.7 | 89.1 | 75.6 | 30.9 | 86.4 | 73.4 |
| Aya-101 | 22.1 | 85.8 | 72.2 | 26.9 | 87.5 | 69.0 | 31.3 | 86.9 | 61.2 | 26.7 | 85.0 | 64.0 |
| Aya-23-8B | 26.0 | 87.9 | 78.8 | 15.7 | 77.3 | 60.0 | 20.8 | 78.3 | 59.7 | 19.7 | 75.1 | 58.5 |
| Aya-23-35B | 26.6 | 87.7 | 78.2 | 22.1 | 82.4 | 67.6 | 28.8 | 83.7 | 67.9 | 25.2 | 79.6 | 64.7 |
| X-ALMA (only SFT) | 29.3 | **88.8** | 80.2 | **35.0** | 90.6 | 80.8 | 47.0 | 91.7 | 80.8 | **39.1** | 89.5 | 78.9 |
| X-ALMA (Ours) | **29.5** | **89.0** | **80.4** | 34.2 | **90.8** | **81.5** | **47.2** | **91.8** | **81.0** | 38.7 | **89.6** | **79.3** |

| Models | af→en BLEU | COMET-22 | XCOMET | da→en BLEU | COMET-22 | XCOMET | de→en BLEU | COMET-22 | XCOMET | is→en BLEU | COMET-22 | XCOMET |
|---|---|---|---|---|---|---|---|---|---|---|---|---|
| LLaMA-3.1-8B-Instruct | 14.5 | 66.2 | 33.9 | 21.0 | 66.5 | 52.5 | 36.0 | 78.6 | 72.3 | 3.2 | 43.4 | 42.0 |
| NLLB-3.3B | 40.6 | 80.3 | 62.7 | 34.4 | 83.0 | 66.0 | 28.6 | 81.3 | 64.5 | 16.2 | 64.2 | 42.9 |
| LLaMAX2-Alpaca-7B | 53.5 | 88.9 | 76.1 | 46.0 | 89.7 | 79.8 | 41.4 | 88.9 | 78.5 | 31.2 | 84.8 | 75.0 |
| LLaMAX3-Alpaca-8B | 53.1 | 89.0 | 76.0 | 45.3 | 89.6 | 79.6 | 40.5 | 88.8 | 78.4 | 32.5 | 85.6 | 75.8 |
| Aya-101 | 43.2 | 86.1 | 65.4 | 42.4 | 89.2 | 75.9 | 39.7 | 88.5 | 77.9 | 27.2 | 82.3 | 68.4 |
| Aya-23-8B | 46.9 | 85.3 | 70.6 | 42.6 | 87.7 | 76.8 | 43.9 | 89.3 | 78.9 | 13.0 | 68.0 | 46.5 |
| Aya-23-35B | 54.3 | 88.3 | 74.9 | 47.3 | 89.7 | 79.4 | 45.1 | 89.5 | 78.6 | 24.5 | 78.5 | 66.1 |
| X-ALMA (only SFT) | **58.8** | 89.9 | 76.2 | 49.6 | 90.2 | 79.5 | **45.7** | 89.6 | 78.7 | **37.7** | 87.1 | 76.3 |
| X-ALMA (Ours) | 58.6 | **90.0** | **76.6** | **49.7** | **90.7** | **80.4** | 45.3 | **89.8** | **79.2** | 37.4 | **87.2** | **76.6** |

| Models | nl→en BLEU | COMET-22 | XCOMET | no→en BLEU | COMET-22 | XCOMET | sv→en BLEU | COMET-22 | XCOMET | Avg. xx→en BLEU | COMET-22 | XCOMET |
|---|---|---|---|---|---|---|---|---|---|---|---|---|
| LLaMA-3.1-8B-Instruct | 27.6 | 78.7 | 74.1 | 23.1 | 72.7 | 54.1 | 36.2 | 75.5 | 70.5 | 23.1 | 68.8 | 57.1 |
| NLLB-3.3B | 25.3 | 81.9 | 68.0 | 32.1 | 80.7 | 63.9 | 35.0 | 82.3 | 64.4 | 30.3 | 79.1 | 61.8 |
| LLaMAX2-Alpaca-7B | 30.4 | 87.1 | 78.7 | 41.5 | 88.5 | 79.3 | 46.0 | 89.8 | 80.1 | 41.4 | 88.2 | 78.2 |
| LLaMAX3-Alpaca-8B | 30.1 | 87.1 | 78.3 | 41.8 | 88.5 | 79.1 | 45.6 | 89.5 | 79.6 | 41.3 | 88.3 | 78.1 |
| Aya-101 | 30.1 | 86.9 | 78.3 | 39.5 | 88.1 | 76.0 | 44.3 | 89.4 | 78.5 | 38.1 | 87.2 | 74.4 |
| Aya-23-8B | 31.9 | 87.5 | 79.0 | 38.5 | 86.5 | 75.8 | 42.6 | 87.9 | 76.7 | 37.0 | 84.6 | 72.0 |
| Aya-23-35B | 33.9 | 87.8 | 78.8 | 43.2 | 88.5 | 78.8 | 46.9 | 89.5 | 79.6 | 42.2 | 87.4 | 76.6 |
| X-ALMA (only SFT) | **34.2** | 87.6 | 78.2 | **45.7** | 89.1 | 79.0 | 50.0 | 90.2 | 79.8 | **46.0** | 89.1 | 78.2 |
| X-ALMA (Ours) | 33.9 | **88.1** | **79.3** | **45.7** | **89.5** | **79.5** | **50.5** | **90.6** | **80.6** | 45.9 | **89.4** | **78.9** |

Table 8: Full results for Group 2 in the Flores test data.

| Models | en→ca | | | en→es | | | en→gl | | | en→it | | |
|---|---|---|---|---|---|---|---|---|---|---|---|---|
| | BLEU | COMET-22 | XCOMET | BLEU | COMET-22 | XCOMET | BLEU | COMET-22 | XCOMET | BLEU | COMET-22 | XCOMET |
| Llama-3.1 | 37.5 | 86.8 | 77.8 | 25.0 | 83.9 | 77.2 | 30.2 | 84.3 | 74.0 | 24.9 | 81.7 | 72.8 |
| NLLB-3.3B | 43.1 | 87.8 | 77.6 | 28.6 | 86.5 | 80.0 | 35.7 | 87.3 | 76.8 | **31.3** | **88.5** | 80.5 |
| LLaMAX2-Alpaca-7B | 37.7 | 87.2 | 78.1 | 25.1 | 85.5 | 79.0 | 31.5 | 86.5 | 77.2 | 26.0 | 87.0 | 78.9 |
| LLaMAX3-Alpaca-8B | 36.3 | 86.5 | 76.5 | 24.1 | 85.0 | 76.1 | 31.2 | 86.4 | 76.6 | 26.5 | 86.9 | 77.7 |
| Aya-101 | 37.8 | 87.1 | 77.9 | 24.2 | 85.3 | 78.2 | 32.7 | 86.7 | 78.0 | 25.6 | 87.0 | 78.5 |
| Aya-23-8B | 25.1 | 81.7 | 71.9 | 27.8 | 86.4 | 80.6 | 17.2 | 82.7 | 77.3 | 30.2 | 88.4 | 81.0 |
| Aya-23-35B | 33.1 | 83.9 | 73.8 | 27.7 | 86.2 | 80.1 | 25.3 | 84.2 | 76.4 | 30.5 | 88.2 | 80.3 |
| X-ALMA (only SFT) | **45.7** | **89.0** | **80.6** | **29.5** | 87.2 | **81.8** | **39.0** | 88.4 | 80.1 | 32.5 | 89.1 | 82.1 |
| X-ALMA (Ours) | 45.3 | **89.0** | **80.6** | **29.5** | 87.3 | 81.0 | 38.8 | 88.7 | 80.6 | 32.7 | 89.3 | 82.3 |

| Models | en→pt | | | en→ro | | | Avg. en→xx | | |
|---|---|---|---|---|---|---|---|---|---|
| | BLEU | COMET-22 | XCOMET | BLEU | COMET-22 | XCOMET | BLEU | COMET-22 | XCOMET |
| Llama-3.1 | 42.4 | 84.2 | 75.7 | 30.5 | 81.2 | 75.1 | 31.7 | 83.7 | 75.4 |
| NLLB-3.3B | 49.6 | 89.6 | 80.4 | 37.6 | 90.2 | 87.1 | 37.6 | 88.3 | 80.4 |
| LLaMAX2-Alpaca-7B | 41.9 | 88.6 | 79.7 | 31.8 | 88.6 | 85.9 | 32.3 | 87.2 | 79.8 |
| LLaMAX3-Alpaca-8B | 41.5 | 88.1 | 77.0 | 32.7 | 88.1 | 84.0 | 32.0 | 86.8 | 78.0 |
| Aya-101 | 32.5 | 85.3 | 60.5 | 34.9 | 89.4 | 86.9 | 31.3 | 86.8 | 76.6 |
| Aya-23-8B | 48.4 | 89.9 | 81.7 | 37.9 | 90.6 | 89.3 | 31.1 | 86.6 | 80.3 |
| Aya-23-35B | 48.6 | 89.7 | 81.0 | 38.4 | 90.7 | 88.9 | 33.9 | 87.1 | 80.1 |
| X-ALMA (only SFT) | 49.9 | 90.2 | 82.4 | 42.2 | 91.5 | 90.6 | 39.8 | 89.2 | 82.9 |
| X-ALMA (Ours) | **50.2** | **90.4** | **82.8** | **43.3** | **91.6** | **90.8** | **40.0** | **89.4** | **83.0** |

| Models | ca→en | | | es→en | | | gl→en | | | it→en | | |
|---|---|---|---|---|---|---|---|---|---|---|---|---|
| | BLEU | COMET-22 | XCOMET | BLEU | COMET-22 | XCOMET | BLEU | COMET-22 | XCOMET | BLEU | COMET-22 | XCOMET |
| Llama-3.1 | 38.7 | 78.1 | 72.7 | 26.6 | 76.2 | 71.4 | 9.2 | 53.2 | 48.5 | 24.1 | 71.6 | 67.3 |
| NLLB-3.3B | 37.9 | 83.7 | 70.6 | 27.1 | 85.3 | 76.3 | 34.7 | 84.0 | 71.7 | 28.8 | 84.4 | 73.1 |
| LLaMAX2-Alpaca-7B | 43.6 | 88.4 | 78.4 | 29.3 | 86.8 | 78.4 | 38.7 | 88.0 | 78.2 | 31.9 | 87.6 | 78.9 |
| LLaMAX3-Alpaca-8B | 42.9 | 88.3 | 78.1 | 29.0 | 86.7 | 78.1 | 38.6 | 88.0 | 78.0 | 31.3 | 87.5 | 78.5 |
| Aya-101 | 41.1 | 87.6 | 75.6 | 28.8 | 86.8 | 78.0 | 35.5 | 86.9 | 72.7 | 31.2 | 87.4 | 78.1 |
| Aya-23-8B | 39.5 | 85.8 | 75.9 | 31.3 | 87.4 | 78.6 | 37.3 | 87.0 | 76.5 | 34.1 | 88.1 | 79.0 |
| Aya-23-35B | 46.3 | 88.4 | 77.8 | 33.1 | 87.7 | 78.5 | 41.7 | 88.5 | 78.0 | 36.0 | 88.3 | 78.9 |
| X-ALMA (only SFT) | 48.6 | 89.2 | 77.8 | **34.9** | 87.7 | 77.8 | **44.9** | 89.0 | 77.9 | **36.9** | 88.3 | 78.5 |
| X-ALMA (Ours) | **48.7** | **89.6** | **78.7** | 33.0 | **87.9** | **79.0** | 44.6 | **89.2** | **78.4** | 35.5 | **88.6** | **79.3** |

| Models | pt→en | | | ro→en | | | Avg. xx→en | | |
|---|---|---|---|---|---|---|---|---|---|
| | BLEU | COMET-22 | XCOMET | BLEU | COMET-22 | XCOMET | BLEU | COMET-22 | XCOMET |
| Llama-3.1 | 43.9 | 81.9 | 74.0 | 20.1 | 64.6 | 61.0 | 27.1 | 70.9 | 65.8 |
| NLLB-3.3B | 42.3 | 86.7 | 75.1 | 31.4 | 83.0 | 74.1 | 33.7 | 84.5 | 73.5 |
| LLaMAX2-Alpaca-7B | 45.7 | 89.0 | 79.9 | 40.0 | 88.8 | 88.2 | 38.2 | 88.1 | 80.3 |
| LLaMAX3-Alpaca-8B | 46.3 | 89.1 | 79.6 | 40.4 | 88.9 | 88.0 | 38.1 | 88.1 | 80.1 |
| Aya-101 | 43.8 | 88.7 | 78.1 | 37.8 | 88.4 | 85.6 | 36.4 | 87.6 | 78.0 |
| Aya-23-8B | 49.7 | 89.7 | 80.0 | 43.5 | 89.5 | 88.6 | 39.2 | 87.9 | 79.8 |
| Aya-23-35B | **51.5** | 89.9 | 80.0 | 46.0 | 89.7 | 88.3 | 42.4 | 88.8 | 80.2 |
| X-ALMA (only SFT) | 51.0 | 89.7 | 79.4 | **46.8** | 89.7 | 88.2 | **43.8** | 88.9 | 79.9 |
| X-ALMA (Ours) | 50.3 | **90.0** | **80.2** | 45.7 | **90.0** | **88.8** | 43.0 | **89.2** | **80.7** |

Table 9: Full results for Group 3 in the Flores test data.

| Models | en→bg | | | en→mk | | | en→ru | | | en→sr | | |
|---|---|---|---|---|---|---|---|---|---|---|---|---|
| | BLEU | COMET-22 | XCOMET | BLEU | COMET-22 | XCOMET | BLEU | COMET-22 | XCOMET | BLEU | COMET-22 | XCOMET |
| Llama-3.1 | 29.6 | 87.8 | 74.2 | 24.9 | 85.7 | 74.3 | 14.4 | 63.0 | 45.1 | 1.4 | 75.3 | 75.4 |
| NLLB-3.3B | 40.5 | 90.9 | 77.5 | 34.4 | 88.8 | 77.4 | 32.2 | 89.2 | 77.5 | 33.8 | 89.0 | 77.1 |
| LLaMAX2-Alpaca-7B | 33.0 | 89.6 | 77.1 | 29.6 | 87.9 | 77.5 | 25.4 | 87.9 | 76.3 | 8.1 | 79.3 | 77.4 |
| LLaMAX3-Alpaca-8B | 32.2 | 89.0 | 76.8 | 29.3 | 87.4 | 77.0 | 26.4 | 87.7 | 76.4 | 5.8 | 76.2 | 73.8 |
| Aya-101 | 34.3 | 90.0 | 78.4 | 30.7 | 88.7 | 79.0 | 27.2 | 88.3 | 77.5 | 23.3 | 82.9 | 73.2 |
| Aya-23-8B | 6.7 | 73.3 | 60.4 | 2.9 | 57.1 | 42.3 | 29.9 | 89.6 | 79.4 | 0.9 | 61.7 | 55.0 |
| Aya-23-35B | 17.0 | 75.7 | 56.3 | 9.6 | 65.4 | 51.3 | 31.2 | 89.6 | 79.1 | 1.1 | 67.4 | 65.1 |
| X-ALMA (only SFT) | **42.1** | 91.7 | 80.9 | 37.3 | 90.4 | 80.9 | 32.3 | 90.1 | 80.2 | 36.4 | 90.2 | 81.4 |
| X-ALMA (Ours) | 41.7 | **91.8** | **81.1** | **37.6** | **90.6** | **81.4** | **32.9** | **90.3** | **80.5** | **36.8** | **90.7** | **81.6** |

| Models | en→uk | | | Avg. en→xx | | |
|---|---|---|---|---|---|---|
| | BLEU | COMET-22 | XCOMET | BLEU | COMET-22 | XCOMET |
| Llama-3.1 | 22.9 | 83.6 | 69.6 | 18.6 | 79.1 | 67.7 |
| NLLB-3.3B | 30.3 | 89.1 | 74.4 | 34.2 | 89.4 | 76.8 |
| LLaMAX2-Alpaca-7B | 24.3 | 88.0 | 74.8 | 24.1 | 86.5 | 76.6 |
| LLaMAX3-Alpaca-8B | 25.5 | 87.9 | 74.3 | 23.8 | 85.7 | 75.6 |
| Aya-101 | 25.1 | 88.7 | 75.7 | 28.1 | 87.7 | 76.8 |
| Aya-23-8B | 29.4 | 90.2 | 78.1 | 13.9 | 74.4 | 63.0 |
| Aya-23-35B | 30.3 | 90.0 | 77.3 | 17.8 | 77.6 | 65.8 |
| X-ALMA (only SFT) | 31.8 | 90.8 | 78.8 | 36.0 | 90.7 | 80.4 |
| X-ALMA (Ours) | **32.0** | **90.9** | **78.9** | **36.2** | **90.9** | **80.7** |

| Models | bg→en | | | mk→en | | | ru→en | | | sr→en | | |
|---|---|---|---|---|---|---|---|---|---|---|---|---|
| | BLEU | COMET-22 | XCOMET | BLEU | COMET-22 | XCOMET | BLEU | COMET-22 | XCOMET | BLEU | COMET-22 | XCOMET |
| Llama-3.1 | 10.8 | 53.7 | 46.7 | 1.5 | 39.7 | 43.1 | 14.4 | 61.0 | 48.2 | 6.0 | 46.6 | 48.8 |
| NLLB-3.3B | 37.6 | 86.0 | 74.1 | 37.1 | 84.3 | 71.6 | 30.7 | 84.2 | 73.1 | 35.8 | 83.4 | 71.1 |
| LLaMAX2-Alpaca-7B | 38.1 | 87.6 | 77.9 | 39.6 | 87.2 | 77.2 | 33.1 | 86.2 | 76.7 | 40.5 | 87.2 | 78.8 |
| LLaMAX3-Alpaca-8B | 38.2 | 87.5 | 77.6 | 39.8 | 87.2 | 77.2 | 33.1 | 86.4 | 76.8 | 40.6 | 87.3 | 78.6 |
| Aya-101 | 32.9 | 85.4 | 70.7 | 33.7 | 84.3 | 70.1 | 32.7 | 86.1 | 76.7 | 35.0 | 85.0 | 72.6 |
| Aya-23-8B | 32.6 | 84.4 | 71.7 | 25.0 | 78.4 | 63.3 | 36.1 | 86.7 | 76.8 | 27.9 | 79.9 | 66.0 |
| Aya-23-35B | 38.2 | 86.7 | 75.5 | 36.2 | 84.6 | 72.4 | 38.6 | 87.1 | 76.7 | 37.8 | 85.3 | 75.0 |
| X-ALMA (only SFT) | **43.4** | 88.4 | 77.9 | 45.6 | 88.2 | 77.1 | **38.7** | 87.0 | 76.6 | **46.2** | 88.4 | 78.8 |
| X-ALMA (Ours) | 42.9 | **88.6** | **78.5** | 45.6 | **88.4** | **77.6** | 36.7 | **87.2** | **77.4** | 44.7 | 88.4 | **79.4** |

| Models | uk→en | | | Avg. xx→en | | |
|---|---|---|---|---|---|---|
| | BLEU | COMET-22 | XCOMET | BLEU | COMET-22 | XCOMET |
| Llama-3.1 | 6.9 | 55.0 | 37.3 | 7.9 | 51.2 | 44.8 |
| NLLB-3.3B | 33.7 | 83.7 | 70.0 | 35.0 | 84.3 | 72.0 |
| LLaMAX2-Alpaca-7B | 36.8 | 86.7 | 75.7 | 37.6 | 87.0 | 77.2 |
| LLaMAX3-Alpaca-8B | 37.0 | 86.8 | 75.7 | 37.7 | 87.0 | 77.2 |
| Aya-101 | 35.5 | 86.2 | 74.9 | 34.0 | 85.4 | 73.0 |
| Aya-23-8B | 40.1 | 87.2 | 75.7 | 32.3 | 83.3 | 70.7 |
| Aya-23-35B | 42.0 | 87.7 | 75.8 | 38.6 | 86.3 | 75.1 |
| X-ALMA (only SFT) | **42.8** | 87.7 | 75.8 | **43.3** | 87.9 | 77.3 |
| X-ALMA (Ours) | 41.0 | **87.9** | **76.5** | 42.2 | **88.1** | **77.9** |

Table 10: Full results for Group 4 in the Flores test data.

| Models | en→fr | | | en→id | | | en→mg | | | en→ms | | |
|---|---|---|---|---|---|---|---|---|---|---|---|---|
| | BLEU | COMET-22 | XCOMET | BLEU | COMET-22 | XCOMET | BLEU | COMET-22 | XCOMET | BLEU | COMET-22 | XCOMET |
| Llama-3.1 | 43.5 | 84.5 | 73.7 | 32.7 | 78.8 | 64.6 | 1.6 | 47.0 | 11.2 | 34.0 | 86.9 | 75.9 |
| NLLB-3.3B | 51.1 | 88.3 | 76.9 | 46.4 | 91.2 | 77.9 | 17.7 | 81.6 | 59.9 | 41.6 | 89.1 | 76.8 |
| LLaMAX2-Alpaca-7B | 42.0 | 86.8 | 75.7 | 38.0 | 89.7 | 77.7 | 4.4 | 64.9 | 33.3 | 35.0 | 88.3 | 76.7 |
| LLaMAX3-Alpaca-8B | 41.2 | 86.4 | 74.6 | 35.6 | 89.0 | 74.1 | 2.4 | 56.8 | 24.4 | 32.5 | 87.4 | 73.7 |
| Aya-101 | 38.3 | 85.3 | 69.5 | 38.7 | 90.0 | 77.7 | 16.1 | 81.1 | 60.8 | 30.7 | 86.3 | 68.3 |
| Aya-23-8B | 48.9 | 88.3 | 77.8 | 42.9 | 91.2 | 80.0 | 0.3 | 31.0 | 4.4 | 22.2 | 87.3 | 79.7 |
| Aya-23-35B | 49.0 | 88.0 | 77.1 | 43.5 | 91.1 | 79.4 | 0.8 | 41.4 | 16.4 | 26.7 | 87.2 | 77.4 |
| X-ALMA (only SFT) | 51.8 | 88.7 | 78.5 | 48.0 | 91.8 | 80.2 | **16.8** | 81.8 | 61.7 | **42.0** | 89.7 | 78.4 |
| X-ALMA (Ours) | **51.9** | **89.0** | **78.9** | 48.2 | **92.3** | **81.2** | 16.1 | **82.1** | **62.4** | 40.9 | **90.2** | **79.7** |

| Models | en→th | | | en→vi | | | Avg. en→xx | | |
|---|---|---|---|---|---|---|---|---|---|
| | BLEU | COMET-22 | XCOMET | BLEU | COMET-22 | XCOMET | BLEU | COMET-22 | XCOMET |
| Llama-3.1 | 3.4 | 73.6 | 53.2 | 37.7 | 87.1 | 74.5 | 25.5 | 76.3 | 58.9 |
| NLLB-3.3B | 5.3 | 84.3 | 71.5 | 41.8 | 88.0 | 75.4 | 34.0 | 87.1 | 73.1 |
| LLaMAX2-Alpaca-7B | 6.0 | 82.5 | 69.7 | 34.9 | 86.7 | 74.6 | 26.7 | 83.1 | 67.9 |
| LLaMAX3-Alpaca-8B | 3.7 | 84.8 | 72.2 | 34.9 | 86.0 | 71.7 | 25.0 | 81.7 | 65.1 |
| Aya-101 | 9.8 | 86.5 | 74.9 | 31.9 | 85.6 | 71.2 | 27.6 | 85.8 | 70.4 |
| Aya-23-8B | 0.7 | 61.0 | 54.0 | 40.3 | 89.0 | 78.1 | 25.9 | 74.6 | 62.3 |
| Aya-23-35B | 6.1 | 63.2 | 39.9 | 40.4 | 89.2 | 77.9 | 27.7 | 76.7 | 61.3 |
| X-ALMA (only SFT) | 11.6 | 87.4 | 76.1 | 43.9 | 89.4 | 78.5 | **36.1** | 88.2 | 75.8 |
| X-ALMA (Ours) | **12.0** | **88.2** | **77.4** | **44.1** | **89.9** | **79.3** | 35.6 | **88.6** | **76.5** |

| Models | fr→en | | | id→en | | | mg→en | | | ms→en | | |
|---|---|---|---|---|---|---|---|---|---|---|---|---|
| | BLEU | COMET-22 | XCOMET | BLEU | COMET-22 | XCOMET | BLEU | COMET-22 | XCOMET | BLEU | COMET-22 | XCOMET |
| Llama-3.1 | 40.1 | 81.6 | 71.6 | 21.9 | 70.6 | 53.2 | 1.8 | 41.6 | 17.1 | 10.8 | 63.4 | 35.7 |
| NLLB-3.3B | 38.1 | 86.6 | 72.7 | 34.3 | 84.5 | 68.5 | 13.5 | 63.3 | 43.5 | 31.4 | 82.1 | 65.3 |
| LLaMAX2-Alpaca-7B | 42.1 | 88.8 | 77.3 | 40.4 | 88.9 | 78.2 | 15.4 | 71.8 | 56.7 | 40.2 | 88.3 | 77.1 |
| LLaMAX3-Alpaca-8B | 41.6 | 88.7 | 76.8 | 40.8 | 89.0 | 78.2 | 19.6 | 76.0 | 60.6 | 41.3 | 88.6 | 77.0 |
| Aya-101 | 41.2 | 88.6 | 77.0 | 38.8 | 88.4 | 75.3 | 27.7 | 79.8 | 61.5 | 39.0 | 87.8 | 73.8 |
| Aya-23-8B | 45.3 | 89.4 | 77.4 | 44.1 | 89.5 | 78.5 | 1.5 | 47.0 | 18.8 | 40.0 | 87.3 | 75.9 |
| Aya-23-35B | 47.0 | 89.5 | 77.0 | 45.7 | 89.8 | 78.4 | 5.3 | 54.1 | 33.0 | 43.9 | 88.7 | 77.0 |
| X-ALMA (only SFT) | **47.8** | 89.6 | 77.3 | **47.3** | 89.6 | 78.2 | **30.1** | 81.9 | 63.3 | **46.9** | 89.1 | 77.0 |
| X-ALMA (Ours) | 46.0 | **89.6** | **77.8** | 45.8 | **90.1** | **78.8** | 29.2 | **81.9** | **63.4** | 45.4 | **89.5** | **77.8** |

| Models | th→en | | | vi→en | | | Avg. xx→en | | |
|---|---|---|---|---|---|---|---|---|---|
| | BLEU | COMET-22 | XCOMET | BLEU | COMET-22 | XCOMET | BLEU | COMET-22 | XCOMET |
| Llama-3.1 | 10.7 | 65.2 | 46.0 | 22.0 | 71.2 | 59.0 | 17.9 | 65.6 | 47.1 |
| NLLB-3.3B | 26.8 | 85.9 | 72.8 | 31.6 | 84.1 | 70.4 | 29.3 | 81.1 | 65.5 |
| LLaMAX2-Alpaca-7B | 21.4 | 81.2 | 71.1 | 32.4 | 86.4 | 75.3 | 32.0 | 84.2 | 72.6 |
| LLaMAX3-Alpaca-8B | 28.2 | 87.7 | 75.4 | 33.7 | 87.2 | 75.9 | 34.2 | 86.2 | 74.0 |
| Aya-101 | 26.9 | 85.8 | 72.1 | 33.6 | 86.6 | 75.7 | 34.5 | 86.2 | 72.6 |
| Aya-23-8B | 15.2 | 78.1 | 62.0 | 37.2 | 87.6 | 76.0 | 30.5 | 79.8 | 64.8 |
| Aya-23-35B | 23.5 | 83.6 | 70.7 | 38.9 | 87.8 | 76.0 | 34.1 | 82.3 | 68.7 |
| X-ALMA (only SFT) | **32.3** | 88.0 | 75.0 | **39.8** | 87.9 | 75.6 | **40.7** | 87.7 | 74.4 |
| X-ALMA (Ours) | 31.7 | **88.6** | **76.2** | 38.2 | **88.2** | **76.5** | 39.4 | **88.0** | **75.1** |

Table 11: Full results for Group 5 in the Flores test data.

| Models | en→cs | | | en→el | | | en→hu | | | en→lt | | |
|---|---|---|---|---|---|---|---|---|---|---|---|---|
| | BLEU | COMET-22 | XCOMET | BLEU | COMET-22 | XCOMET | BLEU | COMET-22 | XCOMET | BLEU | COMET-22 | XCOMET |
| Llama-3.1 | 27.1 | 88.0 | 73.5 | 19.7 | 82.3 | 70.4 | 19.8 | 82.2 | 70.4 | 13.0 | 77.2 | 60.6 |
| NLLB-3.3B | 32.2 | 91.0 | 77.7 | 27.4 | 89.0 | 76.6 | 26.4 | 89.3 | 78.7 | 25.2 | 89.3 | 77.3 |
| LLaMAX2-Alpaca-7B | 26.0 | 89.1 | 75.3 | 19.9 | 86.4 | 74.5 | 19.1 | 87.0 | 74.9 | 19.0 | 87.0 | 74.0 |
| LLaMAX3-Alpaca-8B | 24.6 | 88.1 | 73.4 | 20.4 | 86.2 | 74.2 | 18.2 | 86.6 | 73.7 | 17.0 | 86.1 | 72.8 |
| Aya-101 | 26.7 | 90.0 | 77.2 | 21.4 | 86.6 | 74.3 | 21.4 | 88.4 | 78.6 | 22.5 | 89.2 | 78.6 |
| Aya-23-8B | 30.5 | 91.1 | 79.1 | 26.1 | 89.5 | 80.1 | 3.6 | 51.7 | 21.4 | 5.4 | 65.4 | 42.0 |
| Aya-23-35B | 32.2 | 91.4 | 79.4 | 27.0 | 89.6 | 80.2 | 10.8 | 77.0 | 57.2 | 14.0 | 82.5 | 68.0 |
| X-ALMA (only SFT) | 33.8 | 91.5 | 79.4 | 27.9 | 89.8 | 80.3 | 27.0 | 90.4 | 82.2 | **28.4** | 91.3 | **81.9** |
| X-ALMA (Ours) | **34.4** | **92.1** | **80.3** | 28.7 | **90.1** | **80.6** | 27.3 | **90.7** | **82.7** | 28.3 | **91.5** | 78.9 |

| Models | en→lv | | | en→pl | | | Avg. en→xx | | |
|---|---|---|---|---|---|---|---|---|---|
| | BLEU | COMET-22 | XCOMET | BLEU | COMET-22 | XCOMET | BLEU | COMET-22 | XCOMET |
| Llama-3.1 | 11.4 | 70.7 | 41.4 | 14.2 | 73.5 | 57.1 | 17.5 | 79.0 | 62.2 |
| NLLB-3.3B | 25.0 | 87.4 | 70.9 | 21.6 | 88.9 | 75.5 | 26.3 | 89.2 | 76.1 |
| LLaMAX2-Alpaca-7B | 22.3 | 86.8 | 70.6 | 18.3 | 87.5 | 73.9 | 20.8 | 87.3 | 73.9 |
| LLaMAX3-Alpaca-8B | 21.1 | 85.8 | 68.9 | 17.2 | 86.7 | 71.9 | 19.8 | 86.6 | 72.5 |
| Aya-101 | 25.0 | 88.6 | 74.8 | 18.3 | 87.6 | 75.0 | 22.6 | 88.4 | 76.4 |
| Aya-23-8B | 1.5 | 36.5 | 7.3 | 20.7 | 89.2 | 77.2 | 14.6 | 70.6 | 51.2 |
| Aya-23-35B | 7.9 | 62.7 | 38.1 | 22.4 | 89.8 | 78.1 | 19.1 | 82.1 | 66.8 |
| X-ALMA (only SFT) | 29.3 | 90.7 | 78.5 | **23.3** | 90.1 | 78.9 | 28.3 | 90.6 | **80.2** |
| X-ALMA (Ours) | **30.8** | **91.1** | **79.3** | 23.2 | **90.4** | **79.3** | **28.8** | **91.0** | 80.2 |

| Models | cs→en | | | el→en | | | hu→en | | | lt→en | | |
|---|---|---|---|---|---|---|---|---|---|---|---|---|
| | BLEU | COMET-22 | XCOMET | BLEU | COMET-22 | XCOMET | BLEU | COMET-22 | XCOMET | BLEU | COMET-22 | XCOMET |
| Llama-3.1 | 22.1 | 64.9 | 64.9 | 10.3 | 57.4 | 43.0 | 8.6 | 54.8 | 52.9 | 3.3 | 49.2 | 41.8 |
| NLLB-3.3B | 29.4 | 80.1 | 59.9 | 33.0 | 86.1 | 75.3 | 14.0 | 70.1 | 43.7 | 12.6 | 67.1 | 38.9 |
| LLaMAX2-Alpaca-7B | 37.6 | 87.9 | 77.8 | 24.5 | 77.2 | 71.5 | 32.2 | 87.5 | 78.2 | 29.7 | 85.1 | 73.6 |
| LLaMAX3-Alpaca-8B | 37.5 | 88.1 | 77.8 | 34.2 | 87.5 | 77.5 | 32.5 | 87.8 | 78.6 | 31.0 | 86.0 | 74.4 |
| Aya-101 | 35.6 | 87.6 | 76.4 | 32.1 | 86.5 | 75.3 | 29.9 | 86.4 | 74.6 | 30.2 | 85.8 | 74.2 |
| Aya-23-8B | 40.7 | 88.5 | 78.1 | 36.1 | 87.8 | 77.8 | 23.0 | 81.1 | 67.2 | 24.6 | 80.6 | 65.6 |
| Aya-23-35B | 42.3 | 88.5 | 77.9 | 39.0 | 88.3 | 78.0 | 32.2 | 86.5 | 76.6 | 32.9 | 85.4 | 73.5 |
| X-ALMA (only SFT) | **43.3** | 89.0 | 78.4 | **38.0** | 87.9 | 77.0 | 37.3 | 88.7 | 78.9 | 35.9 | 87.1 | 75.1 |
| X-ALMA (Ours) | 42.6 | **89.2** | **78.8** | 37.4 | **88.3** | **78.3** | **37.6** | **89.1** | **79.7** | **36.7** | **87.4** | **75.7** |

| Models | lv→en | | | pl→en | | | Avg. xx→en | | |
|---|---|---|---|---|---|---|---|---|---|
| | BLEU | COMET-22 | XCOMET | BLEU | COMET-22 | XCOMET | BLEU | COMET-22 | XCOMET |
| Llama-3.1 | 2.6 | 41.7 | 43.2 | 12.8 | 61.1 | 49.6 | 9.9 | 54.8 | 49.2 |
| NLLB-3.3B | 10.4 | 68.1 | 40.9 | 20.3 | 77.8 | 55.4 | 20.0 | 74.9 | 52.4 |
| LLaMAX2-Alpaca-7B | 31.6 | 86.3 | 76.2 | 28.0 | 85.7 | 70.4 | 30.6 | 85.0 | 74.6 |
| LLaMAX3-Alpaca-8B | 32.7 | 87.0 | 77.1 | 28.3 | 85.6 | 70.4 | 32.7 | 87.0 | 76.0 |
| Aya-101 | 32.0 | 86.3 | 74.5 | 28.0 | 85.6 | 70.2 | 31.3 | 86.3 | 74.2 |
| Aya-23-8B | 14.1 | 73.4 | 51.9 | 30.5 | 86.1 | 70.7 | 28.1 | 82.9 | 68.5 |
| Aya-23-35B | 29.1 | 83.3 | 70.3 | 33.4 | 86.7 | 70.8 | 34.8 | 86.4 | 74.5 |
| X-ALMA (only SFT) | **38.2** | 87.9 | 77.2 | **32.8** | 86.5 | 70.5 | **37.6** | 87.9 | 76.2 |
| X-ALMA (Ours) | 37.5 | **88.3** | **78.1** | 32.2 | **86.9** | **71.3** | 37.3 | **88.2** | **77.0** |

Table 12: Full results for Group 6 in the Flores test data.

| Models | en→et | | | en→fi | | | en→ja | | | en→ka | | |
|---|---|---|---|---|---|---|---|---|---|---|---|---|
| | BLEU | COMET-22 | XCOMET | BLEU | COMET-22 | XCOMET | BLEU | COMET-22 | XCOMET | BLEU | COMET-22 | XCOMET |
| Llama-3.1 | 9.3 | 66.3 | 48.1 | 16.0 | 82.0 | 70.4 | 10.8 | 66.0 | 25.0 | 7.0 | 73.1 | 48.1 |
| NLLB-3.3B | 25.0 | 90.5 | 79.7 | 24.1 | 91.7 | 81.1 | 22.6 | 87.9 | 75.1 | 14.8 | 84.6 | 70.2 |
| LLaMAX2-Alpaca-7B | 19.0 | 88.4 | 76.1 | 18.4 | 90.2 | 79.6 | 28.1 | 88.9 | 78.1 | 10.7 | 83.0 | 68.2 |
| LLaMAX3-Alpaca-8B | 18.1 | 87.7 | 75.1 | 17.5 | 89.3 | 77.6 | 27.5 | 89.0 | 77.3 | 9.6 | 78.6 | 55.9 |
| Aya-101 | 21.9 | 90.7 | 81.3 | 18.9 | 90.3 | 78.9 | 27.3 | 89.0 | 77.4 | 11.3 | 85.3 | 72.1 |
| Aya-23-8B | 1.5 | 40.5 | 12.9 | 2.4 | 51.9 | 21.8 | 30.7 | 90.8 | 80.3 | 0.4 | 43.3 | 32.8 |
| Aya-23-35B | 6.1 | 57.8 | 33.0 | 8.1 | 70.0 | 46.3 | 30.9 | 91.0 | 80.3 | 2.0 | 47.6 | 19.0 |
| X-ALMA (only SFT) | 26.4 | 91.6 | 82.6 | 25.3 | 92.7 | 84.6 | 34.6 | 91.2 | 81.0 | 14.0 | 87.6 | 75.7 |
| X-ALMA (Ours) | **27.9** | **92.2** | **84.0** | **26.4** | **92.9** | **85.2** | **36.6** | **91.7** | **81.9** | **15.2** | **88.5** | **76.9** |

| Models | en→ko | | | en→zh | | | Avg. en→xx | | |
|---|---|---|---|---|---|---|---|---|---|
| | BLEU | COMET-22 | XCOMET | BLEU | COMET-22 | XCOMET | BLEU | COMET-22 | XCOMET |
| Llama-3.1 | 6.8 | 71.5 | 41.7 | 14.0 | 67.7 | 37.6 | 10.6 | 71.1 | 45.1 |
| NLLB-3.3B | 12.5 | 88.4 | 79.1 | 32.4 | 82.0 | 64.1 | 21.9 | 87.5 | 74.9 |
| LLaMAX2-Alpaca-7B | 10.3 | 86.8 | 76.7 | 35.2 | 85.5 | 74.0 | 20.3 | 87.1 | 75.4 |
| LLaMAX3-Alpaca-8B | 8.8 | 85.6 | 74.0 | 36.3 | 85.6 | 73.3 | 19.6 | 86.0 | 72.2 |
| Aya-101 | 10.2 | 87.4 | 77.2 | 27.3 | 82.4 | 64.7 | 19.5 | 87.5 | 75.3 |
| Aya-23-8B | 13.1 | 89.0 | 79.5 | 40.2 | 87.3 | 76.8 | 14.7 | 67.1 | 50.7 |
| Aya-23-35B | 12.8 | 89.4 | 80.3 | 37.3 | 87.5 | 77.0 | 16.2 | 73.9 | 56.0 |
| X-ALMA (only SFT) | 15.0 | 89.3 | 80.7 | 43.6 | 88.2 | 77.4 | 26.5 | 90.1 | 80.3 |
| X-ALMA (Ours) | **15.8** | **89.9** | **81.6** | **44.9** | **88.7** | **78.4** | **27.8** | **90.6** | **81.3** |

| Models | et→en | | | fi→en | | | ja→en | | | ka→en | | |
|---|---|---|---|---|---|---|---|---|---|---|---|---|
| | BLEU | COMET-22 | XCOMET | BLEU | COMET-22 | XCOMET | BLEU | COMET-22 | XCOMET | BLEU | COMET-22 | XCOMET |
| Llama-3.1 | 4.1 | 45.9 | 51.6 | 5.8 | 53.3 | 52.8 | 17.7 | 69.4 | 67.6 | 0.3 | 39.2 | 34.3 |
| NLLB-3.3B | 7.2 | 62.5 | 29.9 | 10.2 | 67.7 | 39.2 | 17.2 | 79.5 | 61.0 | 25.6 | 84.8 | 70.3 |
| LLaMAX2-Alpaca-7B | 32.1 | 87.4 | 80.7 | 31.7 | 89.0 | 78.9 | 23.4 | 87.1 | 76.4 | 18.1 | 76.7 | 66.8 |
| LLaMAX3-Alpaca-8B | 33.6 | 88.3 | 82.2 | 31.6 | 89.3 | 79.4 | 24.6 | 87.5 | 76.9 | 1.2 | 50.7 | 51.5 |
| Aya-101 | 32.5 | 87.7 | 80.2 | 29.7 | 88.6 | 77.8 | 23.5 | 86.5 | 75.5 | 25.6 | 84.5 | 69.7 |
| Aya-23-8B | 15.4 | 74.9 | 55.9 | 20.4 | 81.3 | 66.3 | 28.1 | 87.9 | 76.8 | 3.6 | 60.1 | 32.9 |
| Aya-23-35B | 28.9 | 84.2 | 74.5 | 29.8 | 87.3 | 76.2 | 30.4 | 88.4 | 77.1 | 19.4 | 79.4 | 65.0 |
| X-ALMA (only SFT) | 38.2 | 89.2 | 82.5 | 36.0 | 90.0 | 79.6 | 28.9 | 88.1 | 77.0 | 28.4 | 86.8 | 71.6 |
| X-ALMA (Ours) | **38.8** | **89.6** | **83.5** | **36.2** | **90.5** | **80.4** | 28.8 | **88.5** | **77.7** | **29.3** | **87.1** | **72.8** |

| Models | ko→en | | | zh→en | | | Avg. xx→en | | |
|---|---|---|---|---|---|---|---|---|---|
| | BLEU | COMET-22 | XCOMET | BLEU | COMET-22 | XCOMET | BLEU | COMET-22 | XCOMET |
| Llama-3.1 | 15.9 | 66.0 | 62.4 | 24.2 | 78.8 | 69.9 | 11.3 | 58.8 | 56.4 |
| NLLB-3.3B | 26.2 | 84.9 | 73.1 | 16.8 | 77.1 | 52.8 | 17.2 | 76.1 | 54.4 |
| LLaMAX2-Alpaca-7B | 25.2 | 86.6 | 76.4 | 25.8 | 86.5 | 77.2 | 26.1 | 85.5 | 76.1 |
| LLaMAX3-Alpaca-8B | 26.3 | 87.5 | 77.1 | 25.9 | 86.6 | 77.3 | 23.9 | 81.7 | 74.1 |
| Aya-101 | 26.5 | 87.0 | 76.9 | 23.1 | 84.5 | 71.8 | 26.8 | 86.5 | 75.3 |
| Aya-23-8B | 29.4 | 88.0 | 77.3 | 29.4 | 87.1 | 77.4 | 21.0 | 79.9 | 64.4 |
| Aya-23-35B | 32.2 | **88.7** | 77.8 | 32.2 | 87.6 | 77.4 | 28.8 | 85.9 | 74.7 |
| X-ALMA (only SFT) | 30.6 | 88.1 | 77.0 | **30.4** | 87.1 | 76.8 | 32.1 | 88.2 | 77.4 |
| X-ALMA (Ours) | **30.8** | **88.7** | **78.1** | 29.6 | **87.6** | **78.1** | **32.2** | **88.7** | **78.4** |

Table 13: Full results for Group 7 in the Flores test data.

| Models | en→gu | | | en→hi | | | en→mr | | | en→ne | | |
|---|---|---|---|---|---|---|---|---|---|---|---|---|
| | BLEU | COMET-22 | XCOMET | BLEU | COMET-22 | XCOMET | BLEU | COMET-22 | XCOMET | BLEU | COMET-22 | XCOMET |
| Llama-3.1 | 11.9 | 81.7 | 55.7 | 21.7 | 69.6 | 53.5 | 7.6 | 62.4 | 49.8 | 4.3 | 59.6 | 47.5 |
| NLLB-3.3B | 24.3 | 87.2 | 66.2 | 34.4 | 80.9 | 67.7 | 17.1 | 74.3 | 65.6 | 16.4 | 76.5 | 76.8 |
| LLaMAX2-Alpaca-7B | 11.0 | 78.9 | 40.6 | 24.8 | 76.9 | 62.0 | 11.1 | 70.7 | 62.9 | 13.8 | 80.8 | 86.8 |
| LLaMAX3-Alpaca-8B | 13.7 | 82.7 | 57.3 | 23.5 | 76.6 | 61.9 | 10.1 | 69.5 | 63.1 | 10.7 | 78.4 | 83.0 |
| Aya-101 | 15.6 | 83.9 | 60.8 | 21.4 | 75.5 | 57.7 | 10.3 | 69.5 | 60.2 | 10.5 | 77.5 | 79.1 |
| Aya-23-8B | 0.4 | 65.7 | 64.2 | 25.0 | 79.3 | 64.9 | 0.9 | 66.7 | 64.5 | 1.5 | 69.2 | 64.4 |
| Aya-23-35B | 1.5 | 62.2 | 51.1 | 26.0 | 79.1 | 65.6 | 1.3 | 61.1 | 56.3 | 1.4 | 68.3 | 64.0 |
| X-ALMA (only SFT) | **25.0** | 88.2 | 67.9 | **34.3** | 81.4 | 67.6 | **18.0** | 75.9 | 68.3 | **21.5** | 84.0 | 89.5 |
| X-ALMA (Ours) | 24.7 | **88.9** | **68.9** | 34.1 | **81.9** | **68.4** | 17.9 | **76.5** | **69.3** | **21.5** | **84.7** | **90.7** |

| Models | en→ur | | | Avg. en→xx | | |
|---|---|---|---|---|---|---|
| | BLEU | COMET-22 | XCOMET | BLEU | COMET-22 | XCOMET |
| Llama-3.1 | 14.9 | 77.0 | 66.2 | 12.1 | 70.1 | 54.5 |
| NLLB-3.3B | 22.9 | 81.3 | 71.9 | 23.0 | 80.1 | 69.6 |
| LLaMAX2-Alpaca-7B | 16.5 | 77.8 | 68.3 | 15.5 | 77.0 | 64.1 |
| LLaMAX3-Alpaca-8B | 13.4 | 75.6 | 64.8 | 14.3 | 76.5 | 66.0 |
| Aya-101 | 13.9 | 74.6 | 58.5 | 14.3 | 76.2 | 63.3 |
| Aya-23-8B | 0.3 | 63.6 | 64.6 | 5.6 | 68.9 | 64.5 |
| Aya-23-35B | 2.4 | 39.1 | 21.2 | 6.5 | 61.9 | 51.6 |
| X-ALMA (only SFT) | 23.8 | 83.5 | 75.0 | 24.5 | **84.2** | 73.6 |
| X-ALMA (Ours) | **24.0** | **84.1** | **75.8** | **24.5** | 83.2 | **74.6** |

| Models | gu→en | | | hi→en | | | mr→en | | | ne→en | | |
|---|---|---|---|---|---|---|---|---|---|---|---|---|
| | BLEU | COMET-22 | XCOMET | BLEU | COMET-22 | XCOMET | BLEU | COMET-22 | XCOMET | BLEU | COMET-22 | XCOMET |
| Llama-3.1 | 1.8 | 46.2 | 39.2 | 9.2 | 53.7 | 35.8 | 3.2 | 45.1 | 31.1 | 1.6 | 45.5 | 43.4 |
| NLLB-3.3B | 42.3 | 90.2 | 73.5 | 38.7 | 88.9 | 68.3 | 34.0 | 87.0 | 70.1 | 38.0 | 89.7 | 90.9 |
| LLaMAX2-Alpaca-7B | 0.0 | 42.7 | 47.9 | 27.3 | 81.7 | 63.4 | 23.5 | 79.9 | 66.0 | 26.0 | 83.4 | 86.6 |
| LLaMAX3-Alpaca-8B | 9.9 | 66.0 | 60.7 | 35.4 | 88.9 | 67.9 | 30.6 | 87.3 | 70.1 | 32.9 | 89.3 | 90.3 |
| Aya-101 | 28.0 | 82.3 | 65.0 | 34.6 | 87.5 | 66.6 | 30.1 | 85.2 | 68.9 | 31.2 | 84.9 | 86.1 |
| Aya-23-8B | 3.4 | 53.6 | 38.6 | 37.6 | 89.1 | 67.9 | 7.5 | 68.9 | 46.2 | 10.0 | 77.0 | 68.1 |
| Aya-23-35B | 8.8 | 63.1 | 53.0 | 40.1 | 89.6 | 68.2 | 18.4 | 79.9 | 59.9 | 23.3 | 84.1 | 81.7 |
| X-ALMA (only SFT) | 40.4 | 90.1 | 72.3 | **43.0** | 89.8 | 67.7 | **37.7** | 88.5 | 70.4 | 41.2 | 90.6 | 90.9 |
| X-ALMA (Ours) | **40.6** | **90.3** | **72.7** | 42.7 | **90.1** | **68.4** | **37.7** | **88.6** | **70.8** | **41.4** | **90.7** | **91.1** |

| Models | ur→en | | | Avg. xx→en | | |
|---|---|---|---|---|---|---|
| | BLEU | COMET-22 | XCOMET | BLEU | COMET-22 | XCOMET |
| Llama-3.1 | 4.9 | 47.2 | 38.4 | 4.1 | 47.6 | 37.6 |
| NLLB-3.3B | 31.6 | 86.0 | 72.6 | 36.9 | 88.3 | 75.1 |
| LLaMAX2-Alpaca-7B | 25.0 | 81.1 | 69.7 | 20.4 | 73.8 | 66.7 |
| LLaMAX3-Alpaca-8B | 30.5 | 86.5 | 73.1 | 27.8 | 83.6 | 72.4 |
| Aya-101 | 28.1 | 83.9 | 69.7 | 30.4 | 84.8 | 71.2 |
| Aya-23-8B | 9.3 | 70.2 | 50.5 | 13.6 | 71.8 | 54.3 |
| Aya-23-35B | 21.1 | 80.2 | 65.4 | 22.3 | 79.4 | 65.6 |
| X-ALMA (only SFT) | **36.4** | 87.7 | 73.2 | 39.7 | 89.3 | 74.9 |
| X-ALMA (Ours) | **36.4** | **88.0** | **74.0** | **39.8** | **89.6** | **75.4** |

Table 14: Full results for Group 8 in the Flores test data.

| Models | en→ar | | | en→az | | | en→fa | | | en→he | | |
|---|---|---|---|---|---|---|---|---|---|---|---|---|
| | BLEU | COMET-22 | XCOMET | BLEU | COMET-22 | XCOMET | BLEU | COMET-22 | XCOMET | BLEU | COMET-22 | XCOMET |
| Llama-3.1 | 18.9 | 82.9 | 70.0 | 6.6 | 69.6 | 50.7 | 21.4 | 86.1 | 77.3 | 20.8 | 85.3 | 73.5 |
| NLLB-3.3B | 27.5 | 86.3 | 75.2 | 14.0 | 86.9 | 76.6 | 22.6 | 86.5 | 77.5 | 30.4 | 87.8 | 76.1 |
| LLaMAX2-Alpaca-7B | 19.7 | 84.9 | 73.7 | 9.4 | 82.0 | 68.2 | 17.8 | 83.7 | 73.4 | 22.0 | 85.1 | 72.7 |
| LLaMAX3-Alpaca-8B | 14.1 | 82.2 | 69.1 | 7.3 | 80.0 | 65.9 | 17.7 | 84.5 | 74.0 | 23.3 | 86.2 | 74.7 |
| Aya-101 | 17.2 | 84.1 | 72.8 | 11.5 | 85.6 | 75.4 | 19.1 | 86.4 | 77.6 | 20.5 | 85.4 | 73.2 |
| Aya-23-8B | 26.5 | 87.3 | 77.3 | 2.0 | 75.5 | 69.1 | 23.2 | 87.7 | 79.4 | 27.0 | 88.3 | 77.7 |
| Aya-23-35B | 27.4 | 87.1 | 76.6 | 3.0 | 67.2 | 54.8 | 23.8 | 87.6 | 79.1 | 28.9 | 88.2 | 77.0 |
| X-ALMA (only SFT) | **29.1** | 87.8 | 77.6 | **14.0** | 88.2 | 79.0 | **28.4** | 88.5 | 80.2 | 32.7 | 89.6 | **79.6** |
| X-ALMA (Ours) | 28.3 | **88.2** | **78.3** | **14.0** | **88.4** | **79.3** | 27.1 | **88.8** | **80.8** | **33.6** | **89.8** | 79.5 |

| Models | en→kk | | | en→ky | | | en→tr | | | en→uz | | |
|---|---|---|---|---|---|---|---|---|---|---|---|---|
| | BLEU | COMET-22 | XCOMET | BLEU | COMET-22 | XCOMET | BLEU | COMET-22 | XCOMET | BLEU | COMET-22 | XCOMET |
| Llama-3.1 | 8.2 | 77.7 | 61.8 | 3.7 | 58.5 | 39.4 | 19.1 | 84.2 | 69.5 | 9.3 | 81.9 | 66.1 |
| NLLB-3.3B | 20.6 | 90.0 | 78.9 | 13.2 | 88.1 | 74.7 | 29.0 | 89.7 | 76.1 | 18.6 | 89.8 | 75.6 |
| LLaMAX2-Alpaca-7B | 13.0 | 86.5 | 74.7 | 8.2 | 84.0 | 71.9 | 16.2 | 85.1 | 69.4 | 10.1 | 85.1 | 69.3 |
| LLaMAX3-Alpaca-8B | 12.7 | 86.0 | 75.1 | 7.9 | 82.9 | 72.5 | 13.8 | 84.2 | 67.5 | 6.8 | 74.5 | 59.1 |
| Aya-101 | 17.2 | 89.0 | 78.3 | 10.4 | 86.6 | 75.6 | 21.1 | 88.3 | 75.0 | 12.0 | 88.6 | 75.6 |
| Aya-23-8B | 1.2 | 71.0 | 77.0 | 1.2 | 62.9 | 68.9 | 23.7 | 88.9 | 75.8 | 0.5 | 46.5 | 26.6 |
| Aya-23-35B | 0.7 | 45.0 | 21.9 | 0.9 | 49.6 | 34.4 | 23.6 | 88.7 | 74.5 | 0.3 | 37.1 | 17.0 |
| X-ALMA (only SFT) | **22.2** | 90.7 | 80.8 | **13.2** | 88.5 | 78.5 | 27.7 | 90.3 | 78.3 | **16.8** | 90.0 | 77.0 |
| X-ALMA (Ours) | 22.0 | **91.1** | **81.4** | 12.8 | **88.8** | **78.8** | **28.4** | **90.5** | **78.6** | 15.5 | **90.1** | **77.3** |

| Models | Avg. en→xx | | |
|---|---|---|---|
| | BLEU | COMET-22 | XCOMET |
| Llama-3.1 | 13.5 | 78.3 | 63.5 |
| NLLB-3.3B | 22.0 | 88.1 | 76.3 |
| LLaMAX2-Alpaca-7B | 14.5 | 84.6 | 71.7 |
| LLaMAX3-Alpaca-8B | 12.9 | 82.6 | 69.7 |
| Aya-101 | 16.1 | 86.8 | 75.4 |
| Aya-23-8B | 13.2 | 76.0 | 69.0 |
| Aya-23-35B | 13.6 | 68.8 | 54.4 |
| X-ALMA (only SFT) | 23.0 | 89.2 | 78.9 |
| X-ALMA (Ours) | **22.7** | **89.4** | **79.3** |

| Models | ar→en | | | az→en | | | fa→en | | | he→en | | |
|---|---|---|---|---|---|---|---|---|---|---|---|---|
| | BLEU | COMET-22 | XCOMET | BLEU | COMET-22 | XCOMET | BLEU | COMET-22 | XCOMET | BLEU | COMET-22 | XCOMET |
| Llama-3.1 | 17.1 | 59.4 | 45.2 | 3.3 | 46.5 | 41.9 | 23.7 | 72.9 | 56.6 | 3.7 | 48.6 | 24.5 |
| NLLB-3.3B | 38.2 | 86.1 | 74.3 | 15.1 | 77.5 | 61.7 | 29.8 | 83.5 | 69.0 | 39.1 | 86.0 | 73.2 |
| LLaMAX2-Alpaca-7B | 34.3 | 84.8 | 74.3 | 18.6 | 84.5 | 73.2 | 29.0 | 84.1 | 73.7 | 36.8 | 85.8 | 74.3 |
| LLaMAX3-Alpaca-8B | 35.1 | 86.8 | 75.6 | 7.9 | 70.0 | 44.2 | 33.1 | 87.6 | 76.2 | 39.5 | 87.2 | 75.6 |
| Aya-101 | 35.0 | 85.8 | 74.7 | 21.5 | 85.2 | 74.2 | 32.8 | 87.3 | 76.1 | 37.9 | 86.4 | 73.3 |
| Aya-23-8B | 41.5 | 87.9 | 76.4 | 10.6 | 75.6 | 57.2 | 36.8 | 87.9 | **76.3** | 43.2 | 88.4 | 76.5 |
| Aya-23-35B | 43.4 | 87.6 | 75.8 | 17.9 | 82.6 | 69.6 | 39.6 | 88.5 | **76.3** | 46.6 | 88.9 | 76.7 |
| X-ALMA (only SFT) | **41.2** | 87.5 | 75.4 | **25.8** | 86.7 | 74.6 | 37.6 | 88.1 | 75.5 | 44.5 | 88.3 | 75.8 |
| X-ALMA (Ours) | 40.9 | **88.0** | **76.5** | 25.5 | **87.0** | **75.6** | **38.0** | **88.8** | 75.6 | 44.2 | **88.9** | **76.8** |

| Models | kk→en | | | ky→en | | | tr→en | | | uz→en | | |
|---|---|---|---|---|---|---|---|---|---|---|---|---|
| | BLEU | COMET-22 | XCOMET | BLEU | COMET-22 | XCOMET | BLEU | COMET-22 | XCOMET | BLEU | COMET-22 | XCOMET |
| Llama-3.1 | 1.9 | 41.5 | 34.3 | 3.0 | 45.1 | 30.6 | 23.0 | 70.2 | 62.3 | 3.4 | 45.5 | 36.0 |
| NLLB-3.3B | 30.2 | 85.0 | 72.7 | 20.1 | 81.6 | 69.7 | 16.8 | 75.3 | 54.0 | 5.3 | 60.7 | 31.5 |
| LLaMAX2-Alpaca-7B | 27.2 | 85.9 | 75.0 | 18.8 | 83.7 | 73.4 | 30.2 | 87.2 | 75.4 | 25.5 | 84.8 | 72.0 |
| LLaMAX3-Alpaca-8B | 29.0 | 86.7 | 75.5 | 20.4 | 84.5 | 74.1 | 33.4 | 88.6 | 76.9 | 27.9 | 86.1 | 73.5 |
| Aya-101 | 29.2 | 86.1 | 74.9 | 20.4 | 83.0 | 72.6 | 33.2 | 88.1 | 76.0 | 28.1 | 84.9 | 71.9 |
| Aya-23-8B | 3.5 | 59.9 | 39.4 | 4.2 | 64.3 | 45.8 | 35.8 | 88.2 | 76.5 | 3.9 | 61.3 | 30.7 |
| Aya-23-35B | 14.2 | 74.0 | 59.8 | 11.3 | 74.3 | 60.5 | 39.3 | 89.6 | 77.7 | 15.3 | 75.9 | 59.9 |
| X-ALMA (only SFT) | 33.5 | 87.8 | 75.8 | 23.5 | 85.4 | 73.9 | 39.9 | 89.6 | 77.5 | 32.2 | 86.9 | 72.8 |
| X-ALMA (Ours) | **34.7** | **88.1** | **76.6** | **24.7** | **85.7** | **74.7** | **40.0** | **89.9** | **78.2** | **33.5** | **87.4** | **74.1** |

| Models | Avg. xx→en | | |
|---|---|---|---|
| | BLEU | COMET-22 | XCOMET |
| Llama-3.1 | 9.9 | 53.7 | 41.4 |
| NLLB-3.3B | 24.3 | 79.5 | 63.3 |
| LLaMAX2-Alpaca-7B | 27.5 | 85.1 | 73.9 |
| LLaMAX3-Alpaca-8B | 28.3 | 84.7 | 71.4 |
| Aya-101 | 29.7 | 85.9 | 74.2 |
| Aya-23-8B | 22.4 | 76.7 | 59.8 |
| Aya-23-35B | 28.4 | 82.7 | 69.6 |
| X-ALMA (only SFT) | 34.8 | 87.5 | 75.1 |
| X-ALMA (Ours) | **35.2** | **88.0** | **76.0** |

Table 15: Full results for all languages in the WMT'23 test data.

| Models | en→de | | | en→zh | | | en→ja | | | en→ru | | |
|---|---|---|---|---|---|---|---|---|---|---|---|---|
| | BLEU | COMET-22 | XCOMET | BLEU | COMET-22 | XCOMET | BLEU | COMET-22 | XCOMET | BLEU | COMET-22 | XCOMET |
| ALMA-R-13B | 30.4 | 84.0 | 68.8 | 32.3 | 85.0 | 71.3 | - | - | - | 22.8 | 85.5 | **74.4** |
| TowerInstruct-7B-v0.2 | 37.9 | 83.1 | 68.1 | 41.9 | 85.6 | 70.2 | - | - | - | 29.2 | 85.3 | 71.1 |
| NLLB-3.3B | 33.5 | 79.7 | 61.0 | 34.8 | 79.6 | 55.6 | 13.8 | 81.6 | 65.7 | 29.1 | 83.8 | 69.9 |
| LLaMAX2-Alpaca-7B | 18.6 | 74.1 | 59.8 | 39.8 | 82.6 | 64.6 | 15.4 | 83.4 | 70.7 | 22.1 | 81.6 | 67.8 |
| LLaMAX3-Alpaca-8B | 20.9 | 73.3 | 55.2 | 34.0 | 81.5 | 59.9 | 11.9 | 81.8 | 66.7 | 23.5 | 81.6 | 67.6 |
| Aya-101 | 25.1 | 75.1 | 52.9 | 25.4 | 78.6 | 52.0 | 14.1 | 84.6 | 72.0 | 22.1 | 83.1 | 69.9 |
| Aya-23-8B | 29.3 | 80.4 | 66.2 | 44.5 | 85.3 | 68.8 | 19.3 | 86.5 | 75.1 | 24.3 | 84.3 | 71.8 |
| Aya-23-35B | 30.7 | 80.7 | 66.6 | 42.8 | 84.6 | 68.1 | 20.6 | 86.4 | 75.1 | 27.5 | 84.7 | 71.7 |
| X-ALMA (only SFT) | **40.9** | 84.1 | 68.6 | 47.5 | 86.1 | 69.6 | 22.3 | 86.8 | 75.4 | **31.5** | 85.9 | 73.4 |
| X-ALMA (Ours) | 39.4 | **84.4** | **69.4** | **47.9** | **86.7** | **71.3** | **22.7** | **87.5** | **77.1** | **31.5** | **86.3** | 74.0 |

| Models | en→uk | | | en→he | | | Avg. en→xx | | |
|---|---|---|---|---|---|---|---|---|---|
| | BLEU | COMET-22 | XCOMET | BLEU | COMET-22 | XCOMET | BLEU | COMET-22 | XCOMET |
| ALMA-R-13B | - | - | - | - | - | - | - | - | - |
| TowerInstruct-7B-v0.2 | - | - | - | - | - | - | - | - | - |
| NLLB-3.3B | 25.5 | 82.8 | 67.6 | 31.4 | 83.6 | 69.1 | 28.0 | 81.8 | 64.8 |
| LLaMAX2-Alpaca-7B | 20.0 | 80.9 | 65.0 | 23.4 | 81.5 | 66.6 | 23.2 | 80.7 | 65.8 |
| LLaMAX3-Alpaca-8B | 19.8 | 80.6 | 64.9 | 24.4 | 82.5 | 68.5 | 22.4 | 80.2 | 63.8 |
| Aya-101 | 19.7 | 82.7 | 67.8 | 19.8 | 82.0 | 67.2 | 21.0 | 81.0 | 63.6 |
| Aya-23-8B | 24.3 | 84.3 | 70.1 | 26.5 | 84.3 | 71.1 | 28.0 | 84.2 | 70.5 |
| Aya-23-35B | 24.9 | 84.0 | 69.6 | 29.3 | 84.1 | 70.5 | 29.3 | 84.1 | 70.3 |
| X-ALMA (only SFT) | **27.4** | 85.3 | 71.6 | **31.4** | 86.1 | 73.7 | 33.5 | 85.7 | 72.0 |
| X-ALMA (Ours) | 28.3 | **85.5** | **72.2** | 32.5 | **86.2** | **74.1** | **33.7** | **86.1** | **73.0** |

| Models | de→en | | | zh→en | | | ja→en | | | ru→en | | |
|---|---|---|---|---|---|---|---|---|---|---|---|---|
| | BLEU | COMET-22 | XCOMET | BLEU | COMET-22 | XCOMET | BLEU | COMET-22 | XCOMET | BLEU | COMET-22 | XCOMET |
| ALMA-R-13B | 42.6 | 85.5 | 69.1 | 23.2 | 80.6 | 70.6 | - | - | - | 33.0 | **83.3** | 72.1 |
| TowerInstruct-7B-v0.2 | 39.8 | 84.6 | **69.3** | 23.9 | 80.5 | 70.4 | - | - | - | 34.1 | 83.1 | 72.1 |
| NLLB-3.3B | 20.1 | 66.6 | 20.1 | 11.4 | 67.8 | 41.6 | 6.8 | 65.8 | 41.5 | 24.4 | 76.7 | 62.2 |
| LLaMAX2-Alpaca-7B | 22.1 | 78.0 | 66.3 | 20.7 | 78.3 | 68.1 | 16.9 | 79.5 | 68.5 | 29.6 | 81.1 | 70.3 |
| LLaMAX3-Alpaca-8B | 25.6 | 79.4 | 66.7 | 22.3 | 79.3 | 69.2 | 17.6 | 80.1 | 69.3 | 29.4 | 81.3 | 70.5 |
| Aya-101 | 34.9 | 81.6 | 65.1 | 13.8 | 73.7 | 55.5 | 13.9 | 77.3 | 63.7 | 28.4 | 81.4 | 70.5 |
| Aya-23-8B | 32.3 | 82.1 | 67.9 | 22.6 | 78.8 | 68.0 | 19.8 | 80.2 | **68.6** | 30.9 | 81.6 | 70.8 |
| Aya-23-35B | 32.7 | 82.3 | 68.3 | 23.5 | 79.7 | 69.5 | 21.3 | 81.6 | **69.8** | 31.7 | **82.2** | 71.1 |
| X-ALMA (only SFT) | **42.5** | 85.3 | 68.9 | **23.8** | 80.3 | 69.9 | 20.4 | 82.4 | 70.2 | 32.8 | 82.4 | 71.5 |
| X-ALMA (Ours) | 41.7 | **85.7** | 69.1 | 25.1 | **80.9** | **71.2** | 21.0 | 82.4 | 71.2 | 32.0 | **83.3** | **72.3** |

| Models | uk→en | | | he→en | | | Avg. xx→en | | |
|---|---|---|---|---|---|---|---|---|---|
| | BLEU | COMET-22 | XCOMET | BLEU | COMET-22 | XCOMET | BLEU | COMET-22 | XCOMET |
| ALMA-R-13B | - | - | - | - | - | - | - | - | - |
| TowerInstruct-7B-v0.2 | - | - | - | - | - | - | - | - | - |
| NLLB-3.3B | 33.1 | 79.0 | 62.7 | 40.8 | 79.9 | 63.2 | 22.8 | 72.6 | 48.5 |
| LLaMAX2-Alpaca-7B | 39.1 | 85.1 | 72.6 | 36.9 | 81.3 | 68.6 | 27.6 | 80.6 | 69.1 |
| LLaMAX3-Alpaca-8B | 37.8 | 84.9 | 72.7 | 40.1 | 83.0 | 69.7 | 28.8 | 81.3 | 69.7 |
| Aya-101 | 34.9 | 84.5 | 72.3 | 33.7 | 82.9 | 68.5 | 26.6 | 80.2 | 65.9 |
| Aya-23-8B | 39.2 | 85.0 | 72.7 | 46.1 | 84.9 | 70.4 | 31.8 | 82.1 | 69.8 |
| Aya-23-35B | 39.1 | 85.7 | 73.3 | **49.0** | **85.9** | 71.4 | 32.9 | 82.9 | 70.5 |
| X-ALMA (only SFT) | 42.5 | 86.4 | 73.5 | 42.2 | 84.4 | 70.3 | 34.0 | 83.4 | 70.7 |
| X-ALMA (Ours) | **42.9** | **86.8** | **74.0** | 44.2 | 85.6 | **71.5** | **34.5** | **84.1** | **71.5** |

## F   EXAMPLES OF OVER-REJECTION

Figure 7 presents examples of over-rejection in translations from Chinese to English. For each source sentence, we provide translations from the reference, ARPO (implemented on CPO), and CPO. The words where ARPO and CPO differ from the reference are color-highlighted: green indicates that the variation does not affect the meaning, while red indicates a potentially negative impact on translation quality. As shown in Figure 7, CPO exhibits more stylistic variations than ARPO across all translation examples. Although most of the stylistic changes introduced by CPO are accurate and do not impair meaning, a small number are detrimental. Excessive changes in style can result in sub-optimal translations, a phenomenon we refer to as 'over-rejection'.

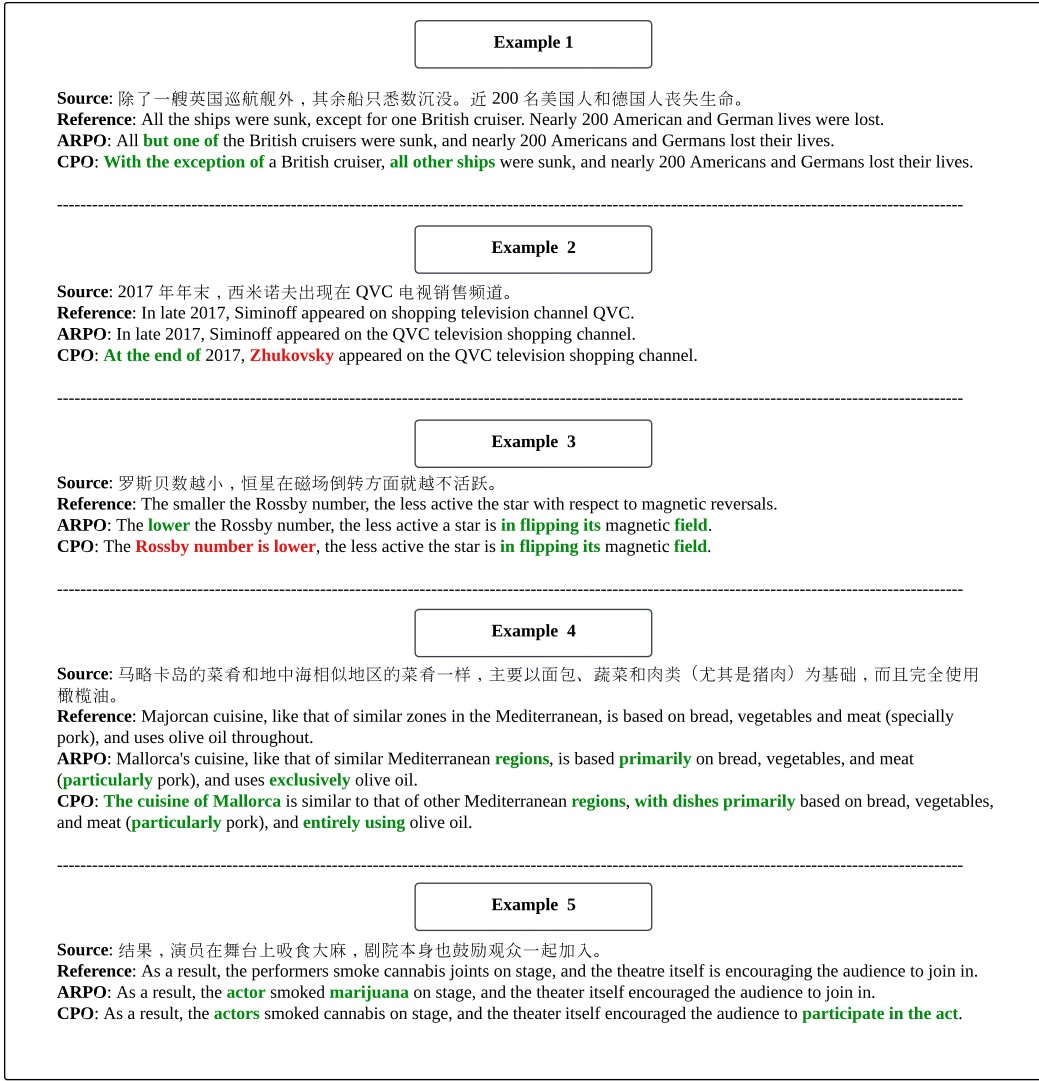

Figure 7:   Examples of over-rejection in Chinese-to-English translation, comparing translations from the reference, ARPO, and CPO. **Green** highlights indicate acceptable variations, while **red** highlights show the potentially harmful changes. CPO introduces more stylistic differences than ARPO, with most being correct but some leading to over-rejection. Although most of variations are correct, the phenomenon of excessive stylistic changes leading to non-optimal translations is referred to as 'over-rejection'.

