# OpenReview forum: "X-ALMA: Plug & Play Modules and Adaptive Rejection for Quality Translation at Scale"
_ICLR.cc/2025/Conference — ICLR 2025 Spotlight_

### Official Review · Reviewer_wi3x · 2024-11-02

**Soundness:** 4
**Presentation:** 4
**Contribution:** 4
**Rating:** 8
**Confidence:** 4

**Summary:**

The paper introduces X-ALMA, a multilingual machine translation model that aims to achieve top-tier performance across 50 diverse languages, regardless of their resource levels. It is friendly to mid- and low-resource languages. The model uses a plug-and-play language-specific module architecture to prevent language conflicts during training and a carefully designed training regimen with novel optimization methods to maximize translation performance. Experimental results demonstrate that X-ALMA outperforms existing multilingual LLMs in every single translation direction on the FLORES-200 and WMT’23 test datasets according to COMET-22. The paper's contributions include the design of X-ALMA's architecture and training methodology.

**Strengths:**

The paper introduces a new approach to multilingual machine translation, featuring a plug-and-play module architecture that effectively addresses language conflicts during training. Additionally, the ARPO method represents a creative combination of existing ideas, aiming to optimize translation performance.
Extensive experimental evaluations demonstrate the effectiveness of the X-ALMA model, achieving superior performance compared to existing open-source multilingual LLMs on multiple language pairs. The paper provides clear descriptions of the background, related work, experimental setup, and analysis of results, making it easy to understand and follow.
This work has the potential to make a significant impact on multilingual machine translation, particularly in terms of delivering high-quality translations for resource-limited languages.
The paper is well-structured and clearly written, providing readers with a comprehensive understanding of the research work.

**Weaknesses:**

The paper presents a comparison with the state-of-the-art in LLM-based machine translation. I am curious about how this compares to the state-of-the-art results of encoder-decoder NMT models that are trained on specific languages.

**Questions:**

On page 5, in Pre-Training Stage 2: For the 10B used in fine-tuning, are an equal number of tokens taken from each language in the group, or are they taken proportionally?

---

> ### Author Response · Authors · 2024-11-17
>
> We sincerely value and appreciate the insightful feedback provided by the reviewer, and we are grateful for the recognition of our technical contributions and analyses.
>
> **Regarding the comparison to language-specific encoder-decoder MT models**
>
> Thank you for your question. Understanding the performance gap between a multilingual model and a dedicated translation model is indeed important. For state-of-the-art encoder-decoder models trained for specific language pairs, we believe a good benchmark is the top models reported in WMT’23 [1]. We compared X-ALMA with the best publicly available encoder-decoder models on the WMT’23 test set, with results presented in the table below. Unlike our approach, which employs a single model for all translation directions, these models **are dedicated to specific language pairs** and different number of parameters. As shown in the table, our method outperforms these language-specific state-of-the-art encoder-decoder models across most languages.
>
> | Models       | de->en | ja->en | ru->en | uk->en | zh->en | he->en |
> |--------------|--------|--------|--------|--------|--------|--------|
> | GTCOM [2]    | 82.7   |        |        | 86.3   |        |        |
> | SKIM [3]     |        | **84.0** |        |        |        |        |
> | PROMPT [4]   |        |        | 80.9   |        |        |        |
> | HW-TSC [5]   |        |        |        |        | 82.8   |        |
> | UvA-LTL [6]  |        |        |        |        |        | 83.3   |
> | X-ALMA       | **85.7** | 82.4   | **83.3** | **86.8** | 80.9   | **85.6** |
>
>
>
> | Models       | en->de | en->ja | en->ru | en->uk | en->zh | en->he |
> |--------------|--------|--------|--------|--------|--------|--------|
> | ZengHui [7]  | 79.4   |        |        |        |        |        |
> | SKIM [3]     |        | 86.6   |        |        |        |        |
> | PROMPT [4]   |        |        | 83.8   |        |        |        |
> | GTCOM [2]    |        |        |        | 82.1   |        |        |
> | HW-TSC [5]   |        |        |        |        | **88.1** |        |
> | UvA-LTL [6]  |        |        |        |        |        | 84.2   |
> | X-ALMA       | **84.4** | **87.5** | **86.3** | **85.5** | 86.7 | **86.2** |
>
> References:
>
> [1]  Findings of the 2023 Conference on Machine Translation (WMT23):  LLMsAreHere But Not Quite There Yet
>
> [2] GTCOMandDLUT’sNeuralMachine Translation Systems for WMT23
>
> [3] SKIM at WMT 2023 General Translation Task
>
> [4] PROMT Systems for WMT23 Shared General Translation Task
>
> [5] Treating General MT Shared Task as a Multi-Domain Adaptation Problem: HW-TSC’s Submission to the WMT23 General MT Shared Task
>
> [6] UvA-MT’s participation in the WMT 2023 general translation shared task.
>
> [7] Achieving State-of-the-Art Multilingual Translation Model with Minimal Data and Parameters

---

> > ### Comment · Reviewer_wi3x · 2024-11-25
> >
> > Thank you for providing the state-of-the-art scores for WMT. Your method performs less effectively in Chinese-English and English-Chinese translation compared to WMT, while it outperforms WMT in other language pairs. Could you explain the reasons for this discrepancy?

---

> > > ### Author Response · Authors · 2024-11-25
> > >
> > > We appreciate the reviewer’s continued discussion on this topic! It is challenging to draw a specific conclusion because the WMT models we reported are designed to support only one language direction, such as HW-TSC, which uses totally different recipe and benefits from additional resources like synthetic data and task-specific methods. In contrast, our approach prioritizes a unified framework that supports all languages. While our model often performs better than other language-specific models, it may not always outperform those that focus significant effort and resources on optimizing a single direction. One notable distinction lies in data usage: we aim for a uniform data framework, relying on datasets like OSCAR and NLLB, rather than optimizing data for individual languages. This suggests that X-ALMA’s performance in specific directions could potentially be improved by incorporating more tailored data for those languages.

---

### Official Review · Reviewer_En5M · 2024-11-02

**Soundness:** 3
**Presentation:** 3
**Contribution:** 3
**Rating:** 6
**Confidence:** 3

**Summary:**

The paper presents plug-and-play adapter modules to improve language scalability for multilingual machine translation with LLMs. Unlike existing models that massively scale the support for new LPs without delivering high quality on all supported languages, the authors claim that using language-specific (LS) modules (for language groups) can reduce interference between languages during training and improve learning and downstream performance. Only the base model and the corresponding language-specific module are activated for a specific LP during inference.

The authors train the complete model (base +LS modules) in three pre-training and two post-training stages. For the last post-training stage, they also propose adaptive rejection to reduce over-rejection, a phenomenon that happens when the preferred and dispreferred translations have high overlap.

Results on FLORES shows that their model achieves consistently high-quality translations across language subgroups as compared to  baselines.

**Strengths:**

1. The authors propose plug-and-play adapters to improve language scalability for LLM-based multilingual machine translation and present a recipe for training such models.
2. The multilingual preference dataset (to be released) can be a valuable resource to the community.
3. Results on the FLORES benchmark show that the proposed approach results in superior quality than baselines across language subgroups, showing the efficacy of their approach.

**Weaknesses:**

1. Using language-specific adapters for multilingual MT is not new.  The paper doesn't establish much connection to prior work that has been done before the LLM era to improve multilingual machine translation by incorporating various modular and adaptable approaches. This is important even if the methodologies and technical setups differ.

2. The ARPO loss introduced in Section 4  doesn't address the core issue which is the granularity at which the preference loss is defined and applied. Instead of diluting the signal from dispreferred response with a penalty when the preferred and dispreferred are close, a more meaningful learning signal would be to help the model focus on differences between the two texts at a finer granularity.  This can be especially important where preferences hinge on subtle but critical nuances of translation. For now, ARPO risks being a temporary fix.

**Questions:**

1. L249-250 Why is parallel text referred to as a pseudo-monolingual dataset?

2. L213-215: what does accurate classification mean here?

---

> ### Author Response · Authors · 2024-11-17
>
> We genuinely appreciate the valuable feedback provided by the reviewer and have addressed them in a point-by-point manner below. We are more than willing to engage in further discussions with the reviewers should any follow-up questions arise.
>
> **Regarding your concern about related works**:
> > The paper doesn't establish much connection to prior work
>
> Thank you for raising legitimate concern.  We fully agree that adding related work on pre-LLM language-specific modules would be beneficial and we will include these in the appendix due to space constraints in the camera-ready version. However, please allow us to show that we have discussed the most relevant works in line 187-190. We kindly request that the reviewer consider the omission of a comprehensive review of language-specific modules —a relatively small part of our paper—as a minor issue.
>
> Furthermore, the core contribution of this paper is to present our training recipe—including architecture, data, training pipeline, and methods—that enables mid- and low-resource languages to achieve practical performance levels. As indicated by Reviewer nWUB, our goal is to provide a timely model and methodology for building a model that performs well across all languages, addressing the longstanding challenge that multilingual models often fail to maintain consistent translation quality when scaling to many languages. Given the numerous aspects we needed to cover, we were unable to include a comprehensive introduction for all previous works.
>
> >Using language-specific adapters for multilingual MT is not new.
>
> We would like to emphasize that the primary focus of our paper is not on the architecture itself but on achieving similar and fair top performance across all considered languages. While we acknowledge that language-specific modules are neither new nor uncommon, our hope is that the relative ease in understanding and reproducing our proposals is why they are most likely to be picked up by the community. We would be happy if the common reaction by the reader was, 'Hunh, this actually works!!'
>
> **Regarding your concern about ARPO**
> >The ARPO loss introduced in Section 4 doesn't address the core issue which is the granularity at which the preference loss is defined and applied.
>
> Thank you for your valuable comment and insightful perspective. We fully agree that employing a more fine-grained signal—such as identifying which specific words should be used or avoided in a translation—would provide a more meaningful learning signal, especially when preferences depend on subtle but critical nuances. This is an excellent idea; however, it necessitates the availability of a preference dataset with fine-grained annotations. If such a dataset were available, not only our method but also more foundational methods like DPO and CPO could leverage this detailed data. Unfortunately, building such a dataset is very expensive and would not be feasible to apply across all preference learning methods. We currently only have labels indicating whether an entire sentence is better or worse, and obtaining fine-grained signals without additional annotation efforts is quite challenging.
>
> **Regarding your question about the term**
> >Why is parallel text referred to as a pseudo-monolingual dataset?
>
> Thank you for your question! Our use of the term "pseudo-monolingual dataset" stems from the intuition that we are pre-training on parallel datasets in a monolingual manner. We wanted to avoid the suggestion that we are explicitly training for MT task at this stage with parallel data. However, we acknowledge that this terminology may cause some confusion and will rephrase this appropriately.
>
> **Regarding your question about “accurate classification”**
>
> > what does accurate classification mean here?
>
> Thank you for your question. By "more accurate classification," we mean that our grouping of languages aligns more closely with human linguistic understanding. This approach ensures that the classified language groups better reflect the actual linguistic relationships and similarities recognized by experts in the field.

---

> > ### Comment · Reviewer_En5M · 2024-11-19
> > **Response to the rebuttal**
> >
> > After reading the rebuttal and reviews from other reviewers, I updated my scores accordingly.
> >
> > I agree that maintaining high-quality translations across language pairs when scaling the number of supported languages is very important, and the paper makes a contribution in that direction. However, I do believe it is important to emphasize the current work's contributions and limitations. The rebuttal clarifies and agrees that the use of adapters is not new, and it's nice that the simple approach works, but the plug-and-play architecture is indeed mentioned as the contribution to this paper (L98).
> >
> > Yes, the suggested loss would require a fine-grained annotation, but it is unclear to me if one could get away from just filtering instances where the edit distance (or any other similarity measure) between the chosen and rejected instances is low. Could you maybe consider showing or elaborating further on how the difference in log-likelihood (example 5) is a better measure for adaptiveness than any other similarity measure?

---

> > > ### Author Response · Authors · 2024-11-19
> > >
> > > Thank you for reading our rebuttal and for raising the score! We greatly appreciate your recognition of our work. We agree that adding more related works is beneficial and will update the paper accordingly.
> > >
> > > Regarding your question:
> > > > Could you maybe consider showing or elaborating further on how the difference in log-likelihood (example 5) is a better measure for adaptiveness than any other similarity measure?
> > >
> > > In Equation 5, we compute the similarity by taking the difference of the average log-likelihoods. This method is both *stable* and *efficient* because the lengths of $y_w$ and $y_l$ are unknown and may vary significantly. By averaging, we control the magnitude of the similarity within a small range, which makes training process *stable*. Additionally, most of the operations involve addition, resulting in minimal computational overhead and make the training *efficient*.
> > >
> > > Essentially, Equation 5 computes similarity between two lists of numbers of unequal lengths. Standard similarity measures, such as geometric distance or correlation metrics (e.g., Pearson correlation coefficient), are not applicable in this scenario due to the different lengths. While there are some methods like dynamic time warping (DTW) being able to handle sequences of different lengths, they require some extra efforts to control the output range and, obviously, they involve higher computational costs. We also avoid using string metrics like edit distance because it may not reflect the sematic difference of the preferred and dispreferred sentence, e.g., the swapping of the first and second sentence leads to large edit distance but the meaning is actually the same (we consider it as high similarity while edit distance does not).

---

### Official Review · Reviewer_nWUB · 2024-11-03

**Soundness:** 3
**Presentation:** 4
**Contribution:** 3
**Rating:** 8
**Confidence:** 5

**Summary:**

The paper presents an approach to multilingual machine translation that focuses on mitigating losses on non-high-resource language pairs. The authors focus on 50 languages ranging from low- to high-resource. The approach consists of performing several levels of continued pretraining and post-training using LoRA modules that group language families together. The training recipe introduces a new preference learning method called ARPO that aims to mitigate "over-rejection" of samples that seems to occur in machine translation for pairs of hypotheses that are very similar to each other in which the differences are just a few characters or tokens. Overall the results indicate state-of-the-art results on multilingual MT benchmarks when compared to other open source multilingual LLMs.

**Strengths:**

The paper is very well written, clear and easy to read. The motivation is solid, timely and important: multilingual LLMs usually do not consistently maintain translation quality when we scale the number of languages. The experimental settings are sound and robust. The preference learning method proposed, ARPO, is very useful and takes into consideration shortcomings of previous methods applied to MT. It is going to be useful for the community working on MT and similar problems.

**Weaknesses:**

* It's unclear if the method works for non English directions. From what I could understand one could load more than one language module and translate across language families. Could you clarify that?
* All the evaluation relies on reference-based automated metrics. There's no human evaluation to validate if the findings automated metrics hold

**Questions:**

Questions made above

---

> ### Author Response · Authors · 2024-11-17
>
> We genuinely appreciate the valuable feedback provided by the reviewer and have addressed them in a point-by-point manner below. We are more than willing to engage in further discussions with the reviewers should any follow-up questions arise.
>
> **Regarding non-English translation**:
> >  It's unclear if the method works for non English directions.
>
> Thank you for raising this question! If we understand correctly, the reviewer is referring to non-English to non-English translation. Our model is primarily designed for English-centric translation, which may lead to some variability in performance when translating directly between non-English languages. This is due to the prompts we used during post-training: “Translate this from <xx> to English” or “Translate this from English to <xx>.” The model may encounter difficulties when faced with an unseen prompt, such as “Translate this from <xx> to <yy>.” To address this, further fine-tuning is necessary. However, if X-ALMA is to be used for non-English to non-English translation, an effective workaround is to first translate from the source language to English, and then from English to the target language, which can yield high-quality results. Direct translation between non-English languages can also be achieved within the same language group. However, challenges persist for direct translation across different language groups because the architecture design of X-ALMA allows one language module to be activated per forward pass to save the memory cost.
>
> **Regarding Evaluation**:
> > All the evaluation relies on reference-based automated metrics.
>
> Thanks for your insightful comment. We totally agree that adding human evaluations can offer more accurate evaluations. However, we don’t have sufficient budget to conduct human evaluation. Alternatively, we respectfully think that the concerns regarding the correlation between human evaluation and automated metrics, such as COMET-22, are largely addressed by the significant performance gap between our models and the baselines. As reported in prior work [1], for instance, an improvement of 0.85 on COMET-22 correlates with a 90% alignment with human evaluation decisions, and our improvements exceed this threshold significantly. Specifically, for Group 1 languages (en→xx), X-ALMA outperforms Aya-101 by an impressive 4.6 points.  Moreover, we would like to clarify that we also consider the reference-free metric XCOMET; however, due to space constraints, these scores were placed in the appendix.
>
> Reference:
>
> [1]  Navigating the metrics maze: Reconciling score magnitudes and accuracies..

---

> > ### Comment · Reviewer_nWUB · 2024-11-21
> >
> > Thank you for the clarification. It would be good if the first point is addressed in the paper maybe as a limitation of the method or at least mentioning it.

---

> > > ### Author Response · Authors · 2024-11-22
> > >
> > > We sincerely appreciate the reviewer's valuable suggestions and active engagement in the discussions. We will definitely add this in the camera-ready version! Thank you!

---

### Meta-Review · Area_Chair_m3yK · 2024-12-18

**Metareview:**

The paper introduces X-ALMA, a multilingual machine translation model designed to enhance translation quality across 50 languages, including low- and mid-resource languages. The approach focuses on mitigating losses in non-high-resource language pairs and preventing language interference during training. This is achieved through plug-and-play language-specific modules that activate only the necessary components for each language pair, thereby reducing interference and improving performance. The training process involves several stages of pretraining and post-training, incorporating a new preference learning method called ARPO, which addresses the issue of "over-rejection" in translation samples with minor differences. Additionally, adaptive rejection techniques are employed to refine translation quality further.

Reviewers made suggestions, including experimentation on non-English directions, comparison with state-of-the-art results of encoder-decoder NMT models, and employing a more fine-grained signal. We appreciate the authors for their efforts in the rebuttal and are pleased to recommend the paper for acceptance.

**Additional Comments On Reviewer Discussion:**

For the non-English translation, the authors acknowledge that X-ALMA is primarily optimized for English-centric translation, which may affect performance when translating directly between non-English languages. They suggest using a two-step translation process—first to English, then to the target language. For the human evaluation, they argue that the significant performance improvements of X-ALMA over baseline models, as measured by automated metrics like COMET-22, suggest a strong correlation with human evaluation. For the connection to prior work that has been done before the LLM era, the authors acknowledge the importance of including related work on pre-LLM language-specific modules and plan to add this information in the appendix due to space constraints. The authors also compare the result to the state-of-the-art results of encoder-decoder NMT models to address the reviewers concerns.

The author has a deep knowledge of machine translation  and actively answers the reviewer's questions.

---

### Decision · Program_Chairs · 2025-01-22

Accept (Spotlight)